# UMFC: Unsupervised Multi-Domain Feature Calibration for Vision-Language Models

Jiachen Liang[1,2], Ruibing Hou[1]*, Minyang Hu[1,2], Hong Chang[1,2], Shiguang Shan[1,2], Xilin Chen[1,2]

[1] Institute of Computing Technology, Chinese Academy of Sciences
[2] University of Chinese Academy of Sciences
{jiachen.liang, minyang.hu}@vipl.ict.ac.cn, {houruibing, changhong, sgshan, xlchen}@ict.ac.cn

## Abstract

Pre-trained vision-language models (*e.g.*, CLIP) have shown powerful zero-shot transfer capabilities. But they still struggle with domain shifts and typically require labeled data to adapt to downstream tasks, which could be costly. In this work, we aim to leverage unlabeled data that naturally spans multiple domains to enhance the transferability of vision-language models. Under this unsupervised multi-domain setting, we have identified inherent model bias within CLIP, notably in its visual and text encoders. Specifically, we observe that CLIP's visual encoder tends to prioritize encoding domain over discriminative category information, meanwhile its text encoder exhibits a preference for domain-relevant classes. To mitigate this model bias, we propose a *training-free* and *label-free* feature calibration method, Unsupervised Multi-domain Feature Calibration (UMFC). UMFC estimates image-level biases from domain-specific features and text-level biases from the direction of domain transition. These biases are subsequently subtracted from original image and text features separately, to render them domain-invariant. We evaluate our method on multiple settings including transductive learning and test-time adaptation. Extensive experiments show that our method outperforms CLIP and performs on par with the state-of-the-arts that need additional annotations or optimization. Our code is available at `https://github.com/GIT-LJc/UMFC`.

## 1 Introduction

Recently, Vision-Language Foundation Models (VLFMs) such as CLIP [30], BLIP [24], Flamingo [1] and ALIGN [20] have demonstrated remarkable performance across various downstream tasks. These VLFMs formulate the training objective as contrastive learning, leveraging millions of image-text pairs to establish a shared embedding space. Equipped with a wide range of visual and text representations, VLFMs exhibit the capability to tackle downstream tasks in a zero-shot manner.

Despite VLFMs being exposed to abundant examples, they may still encounter examples with new variations in downstream tasks. To address the problem of distribution shift between the pre-training and downstream domains, a natural approach involves fine-tuning VLFMs on various target tasks, such as prompt engineering [39, 38] and adapter learning [11, 37]. However, these methods generally require labeled samples for fine-tuning, which is prohibitively expensive to be satisfied in reality. Conversely, abundant unlabeled data are often available for downstream tasks. Notably, in practical scenarios, the unlabeled data typically contains multiple domains, which exacerbates the adaptation difficulty of VLFMs. Therefore, in this paper, we aim to improve the adaptation performance of VLFMs directly on unlabeled data that naturally spans multiple domains.

---

*Corresponding author

38th Conference on Neural Information Processing Systems (NeurIPS 2024).

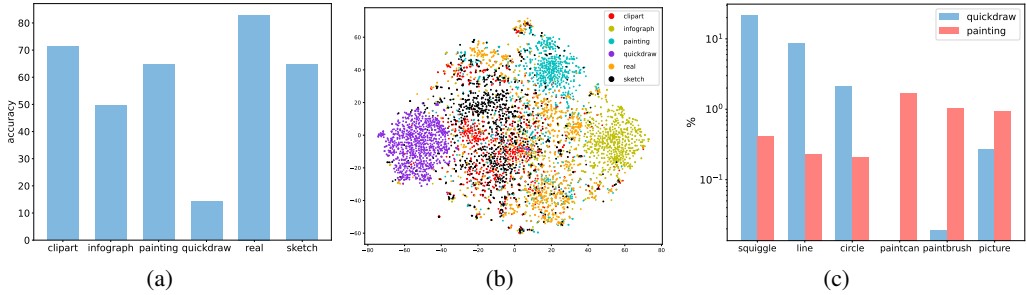

Figure 1: On DomainNet dataset, we visualize (a) The accuracy of CLIP on the six domains. (b) The image features extracted by CLIP's image encoder across different domains. The visualization show that CLIP exhibits inherent model bias. (c) The number of predictions for different classes on *quickdraw* and *painting* domains.

In this unsupervised multi-domain setting, we observe that CLIP cannot perform well when unlabeled data are drawn from mixed distributions. As shown in Figure 1(a), even within the same class space, the accuracy of CLIP varies significantly across different domains. While CLIP performs exceptionally well for images from common distributions encountered during pre-training, *e.g.*, achieving $83.0\%$ accuracy on *real* domain, it struggles with rare distributions encountered during pre-training, *e.g.*, only achieving $14.2\%$ accuracy on *quickdraw* domain. Above observations highlight that CLIP exhibits *model biases* that lead to incorrect predictions in specific scenarios. This raises a fundamental question: *where do these biases originate*?

We point out that these model biases stem from deficiencies in the visual encoder and textual encoder. *On the side of visual encoder, we observe that CLIP' visual encoder prioritizes encoding domain information over discriminative category information.* As shown in Figure 1(b), features from the same domain clearly cluster together, whereas a notable gap separates features from different domains. This phenomenon indicates that CLIP's visual encoder exhibits a higher sensitivity to domain information over category information. When the domain of downstream tasks shifts from pre-trained tasks, the mismatch in domain-specific information encoded in image features could adversely affect classification accuracy. *On the side of textual encoder, we observe that CLIP demonstrates varying category preferences across different domains.* Specifically, CLIP tends to classify images into categories whose name are closely related to corresponding domain. As shown in Figure 1(c), within "quickdraw" domain, a large portion of samples ($\sim 30\%$) are classified as "squiggle" or "line" categories. Conversely, within "painting" domain, CLIP favors categories like "paintcan" and "paintbrush". This observation shows that the class embeddings encoded by CLIP's textual encoder inherently carry domain-specific information, misleading the model to prioritize categories highly associated with respective domains. Due to the combined effects of visual and textual encoder biases, CLIP's performances vary significantly across different domains.

In this paper, we aim to calibrate CLIP to mitigate the model biases. Initially, we analyze the model biases from a probabilistic standpoint. The factors influencing $p(y|x)$, affected by domain variable $z$, can be decoupled into two parts: the sample distribution conditioned on classes $p(x|y, z)$ and the class distribution $p(y|z)$. If the two probability distributions are independent of $z$, domain shifts will not affect the predictions. To this end, we propose **Unsupervised Multi-domain Feature Calibration** (UMFC), a simple yet efficient framework for calibrating CLIP to generalize to various downstream tasks using multi-domain unlabeled data. UMFC jointly calibrates CLIP through two *training-free* modules: Image Feature Calibration module (IFC) and Text Feature Calibration module (TFC). **Firstly**, IFC focuses on calibrating CLIP's image encoder to prioritize category-level over domain-level information, thereby reducing prediction error caused by domain shifts. Specifically, we calculate the average image features for each domain $i$, denoted as $\mu_i$, and posit that the prediction of $\mu_i$ reflects the inherent bias of CLIP's image encoder within that domain. By subtracting this domain-specific bias from original predictions, we can derive domain-agnostic predictions. **Secondly**, TFC focuses on calibrating CLIP's text encoder to remove its preference towards domain-related class names. As observed by [7, 28, 8], a "global direction" exists in CLIP, representing the shift from training distribution to unseen distribution, shared across image and text embedding spaces. Motivated by this observation, we utilize the shift direction between different-domain images to estimate the text shift direction. Then, we subtract this shift vector to counteract the text encoder's bias towards domain-related categories. By combing IFC and TFC, we can calibrate the features on both CLIP's image and text encoders, thereby alleviating the model bias in downstream tasks.

We validate the efficacy of UMFC on three downstream tasks: unsupervised calibration, transductive learning, and test-time adaptation, demonstrating consistent gains over CLIP. UMFC presents a low-cost solution for classification, unlocking the potential of CLIP-like models for practical scenarios characterized by abundant images across multiple domains but scarce labels.

## 2 Related Work

**Few-Shot Learning.** Few-shot learning (FSL) [17, 16, 18, 10, 34] aims to learn a good model on a novel task with few labeled samples. Traditional FSL methods [17, 16, 18, 10, 34] often rely on abundant related training tasks for pre-training and design specialized algorithms to transfer cross-task knowledge, facilitating task adaptation. Additionally, some semi-supervised learning (SSL) methods [26, 14, 15] tackle the scarcity of labeled samples by assuming access to extensive unlabeled data. While both FSL and SSL methods have achieved encouraging results, their generalization abilities are limited. Recently, the CLIP model is proposed, which is a vision-language foundation model that learns a shared vision-language embedding space. As pre-trained on large-scale data, CLIP exhibits impressive zero-shot ability across various downstream tasks. Based on CLIP model, many works focus on improving its performance further on downstream tasks with few labeled data. For example, CoOp [39], CoCoOp [38], MaPLe [21] and PromptSRC [22] leverage few labeled samples to learn the prompt in the continual input embedding space, offering a parameter-efficient way of fine-tuning foundation models. Similarly, CLIP-Adapter [11] and Tip-Adapter-F [37] introduce a lightweight adapter module to produce adaptive multi-modal features. However, these methods require extra labeled data and the process of tuning pre-trained parameters for CLIP, which is cost-expensive. Differently, in this work, we focus on utilizing unlabeled data to enhance CLIP's performance in a training-free manner.

**Domain Adaptation / Domain Generalization.** Recently, several methods [23, 3, 8, 31, 2, 33, 5, 19, 7] exploit a large-scale pre-trained model (*e.g.*, CLIP) to address the domain adaptation and generalization problems. For example, RISE [19] leverages CLIP as a teacher to regularize the student's learned representation through images. The work [3] utilizes domain-invariant and domain-specific prompts for multi-source unsupervised domain adaptation. Other works [8, 7, 5] focus on utilizing the transferability between visual and textual modalities to guide domain information transfer. PromptStyler [5] attempts to simulate various distribution shifts to explore diverse styles in a joint vision-language space. PODA [8] and LADS [7] generate samples in the style of the target domain, and adapt to the target domain based on these samples. However, these language-guided methods require the prior of target-domain names, which may not be satisfied in reality. In contrast, our UMFC method does not require any extra information about the target domain.

**Test-Time Adaptation.** Test-time adaptation aims to adapt a pre-trained model to test tasks, where distribution of test data differs from that of pre-training data. TPT [32] proposes a test-time prompt tuning strategy, which extends traditional TTA methods to vision-language models. Building upon TPT, DiffTPT [9] utilizes pre-trained diffusion models to augment the diversity of test data samples used in prompt tuning. SwapPrompt [27] employs a dual prompts paradigm to enhance the swapped prediction mechanism. However, these prompt learning methods are computationally expensive and time-consuming. Different from above methods, our UMFC only needs to calibrate features in a training-free way, making it more efficient for test-time adaptation.

## 3 CLIP and Model Biases

In this section, we first describe the backgrounds of CLIP and then analyze its bias issue in downstream tasks. We attribute the cause of bias to the visual encoder bias and text encoder bias.

### 3.1 Contrastive Language-Image Pre-training (CLIP)

CLIP [30] consists of two parallel encoders: a visual encoder that maps image inputs into image features, and a text encoder that maps text inputs into text features. The model is trained with a contrastive loss that maximizes similarity between positive image-text pairs while minimizing similarity between negative pairs. This process ensures alignment between the features of images and their corresponding textual descriptions within the feature space. Trained on a vast collection

of image-text pairs, CLIP benefits from general visual representations, endowing it with powerful zero-shot capabilities.

Formally, we denote a CLIP model as $\{F, T\}$, with $F$ and $T$ being the visual and text encoders. Considering a $K$-class classification problem, we use $\mathcal{Y} = \{y_1, \ldots, y_K\}$ to represent the class space, where $y_k$ denotes the class name of $k^{th}$ class. In the zero-shot inference phase, CLIP uses hand-crafted prompts p (such as "a photo of a { }") to covert each class name $y_i$ to category-specific text description $\{p; y_i\}$. Then, we feed these class descriptions to the text encoder to get the text features $\{t_1, \ldots, t_K\}$, where $t_i = T(p; y_i)$. Meanwhile, given a test image $x$, the visual encoder is used to compute its visual feature, denoted as $f = F(x)$. The prediction probability on $x$ can be then computed as:

$$p(y_i|x) = \frac{\exp(\mathrm{sim}(f, t_i)/\tau)}{\sum_{j=1}^{K} \exp(\mathrm{sim}(f, t_j)/\tau)}, \tag{1}$$

where sim denotes the cosine similarity and $\tau$ is the temperature of the softmax function.

## 3.2 Model Bias in CLIP

As shown in Figure 1(a), the accuracy of CLIP varies significantly across different domains within the same class space, *e.g.*, $83.0\%$ in *real* domain and $14.2\%$ in *quickdraw* domain. This phenomenon indicates that CLIP favors domains commonly encountered during pre-training (such as natural images). We attribute this model bias to visual encoder bias and text encoder bias.

- **Visual Encoder Bias.** We observe that CLIP' visual encoder prioritizes encoding domain information over discriminative category information, as shown in Figure 1(b). Due to the abundance of natural images in pre-training, CLIP's visual encoder and text encoder are well-aligned in the natural image domain. Consequently, when the style of input images largely deviates from natural domains (such as *quickdraw* and *infograph* style), the visual feature gap across different domain (as depicted in Figure 1(b)) hinders the text encoder's capacity to process these shifted images effectively, leading a significant drop in classification accuracy.

- **Text Encoder Bias.** We observe that CLIP exhibits a preference for domain-related categories in specific domains, as shown in Figure 1(c). In domains characterized by distinct styles, certain category names may carry domain-specific information. For instance, in *quickdraw* domain, where most images consist of lines and squiggle, CLIP demonstrates a severe bias towards the "line" and "squiggle" categories. This leads to a large number of quickdraw samples being incorrectly classified into the two categories. Conversely, in the *painting* domain, as all images inherently possess painting features, CLIP shows a strong preference for categories related to the concept of *painting*, such as "paintcan" and "paintbrush".

# 4 Unsupervised Multi-domain Feature Calibration

In this section, we continue to analyze the model biases issue from a probabilistic view. Then we give a detailed introduction of our method aimed at alleviating CLIP's biases.

## 4.1 Analyze Model Biases in a Probabilistic View

We start from the posterior probability $p(y_i|x)$, as introduced in Equation 1. Recall that our goal is to maximize the posterior probability $p(y_i|x)$ for each image-label pair $(x, y_i)$ of any domain. A natural question arises: *How different domains affect the posterior probability?* We answer this question from a probabilistic view. Based on the Bayes' Theorem, we can drive that:

$$p(y_i|x) = \frac{p(y_i, x)}{p(x)} = \frac{\sum_z p(y_i, x, z)}{p(x)}, \tag{2}$$

where $z$ is a latent variable that denotes the domain labels. We can further decompose the joint probability $p(y_i, x, z)$ in Equation 2 as $p(y_i, x, z) = p(x|y_i, z) p(y_i|z) p(z)$. Assume that the probability distribution of domains $p(z)$ is uniform, thus maximizing the posterior probability

$p(y_i|x)$ is equal to maximize the summation $\sum_z p(x|y_i, z) p(y_i|z)$, as

$$\max p(y_i|x) = \max \frac{\sum_z p(x|y_i, z) p(y_i|z) p(z)}{p(x)}$$
$$= \max \sum_z p(x|y_i, z) p(y_i|z). \tag{3}$$

Equation 3 shows that the domains affect posterior probability distribution $p(y|x)$ by disturbing two terms: the sample distribution conditioned on classes $p(x|y, z)$ and the class distribution $p(y|z)$.

## 4.2 UMFC: Unsupervised Multi-domain Feature Calibration

As analyzed in Section 3.2, CLIP exhibits both visual and text encoder bias, impacting its generalization ability to downstream tasks. As shown in Equation 3, these biases stem from the probability distributions $p(x|y, z)$ and $p(y|z)$, which are disturbed by domain information. To mitigate the model biases, an intuitive idea is to make the two probability distributions $p(x|y)$ and $p(y)$ independent of domain variable $z$ by calibrating image and text features. To this end, we propose *training-free* UMFC approach, consisting of an Image Feature Calibration (IFC) module to alleviate visual encoder bias and Text Feature Calibration (TFC) module to alleviate text encoder bias, facilitating the transfer of CLIP to downstream tasks. Algorithms are provided in the Appendix C.

**Image Feature Calibration Module.** On the side of visual encoder, we focus on making the conditional probability $p(x|y)$ independent of domain variable $z$, which is equivalent to achieving $p(x|y, z) = p(x|y)$. A straightforward method is to align image feature distributions given a class $y$ across different domains. Unfortunately, as only mixed unlabeled data is provided, we cannot access to class labels and domain labels of images. However, we empirically observe that CLIP's visual encoder prioritizes encoding domain information over discriminative category information. Thus, we can distinguish image features from different domains through a simple clustering algorithm. Formally, we assume that there are $M$ clusters $\{c_1, ..., c_M\}$ after applying a clustering algorithm. Each cluster is assumed to follow a Gaussian distribution $c_i \sim \mathcal{N}(\mu_i, \Sigma_i)$, where mean vector $\mu_i = (\frac{1}{|c_i|} \sum_{f \in c_i} f)$ represents the average of image features from cluster $c_i$. Due to the model biases, the pseudo labels produced by zero-shot CLIP are not reliable. Therefore, we directly align the margin image feature distribution of each cluster by subtracting the corresponding mean vector[2]. Specifically, for each visual feature $f$ belonging to the cluster $c_i$, we calibrate it as follows:

$$f' = \frac{f - \mu_i}{\|f - \mu_i\|_2}. \tag{4}$$

**Text Feature Calibration Module.** On the side of text encoder, we focus on making the class probability $p(y)$ independent of domain variable $z$, which is equivalent to achieving $p(y|z) = p(y)$. Previous works [35] have found that CLIP's performance is sensitive to class name $y$. For example, replacing category names with synonyms or near-synonymous terms can significantly impact CLIP's prediction results. Due to the influence of domain information in image features, text encoder bias can cause CLIP to categorize domain-specific images into categories whose names are semantically similar to that domain. We further observe that such sensitivity to class names varies across different domains, as shown in Figure 1(c). This observation inspires us to calibrate text features by removing domain-specific information. However, a challenge arises as domain labels are unavailable. To address this issue, we attempt to extract domain information from unlabeled images, and then transfer this domain information to the text embedding space to estimate the text bias.

**Observation 4.1** *Cross-Modality Transition Direction. The underlying assumption behind using images to simulate the corresponding shifts in texts is that the transition direction from domain $i$ to domain $j$ is consistent across both the image embedding and text embedding spaces [7, 28, 8], which can be formulated as:*

$$\frac{F(x^i) - F(x^j)}{\|F(x^i) - F(x^j)\|_2} \approx \frac{T(\mathrm{p}^i; y^i) - T(\mathrm{p}^j; y^j)}{\|T(\mathrm{p}^i; y^i) - T(\mathrm{p}^j; y^j)\|_2}, \tag{5}$$

---

[2]This operation is based on the assumption of a uniform class distribution in each cluster. We experimentally found that this operation remains effective, even if this assumption does not hold strictly.

where $(x^i, y^i)$ and $(x^j, y^j)$ represent training samples from domain $i$ and domain $j$ respectively, $p^i$ and $p^j$ denote the domain-specific text prompts for domain $i$ and $j$ respectively. For example, the text prompt of "quickdraw" domain can be "a quickdraw image of a [class]".

Inspired by Observation 4.1, we estimate the text-level domain transition direction using different-domain images. By clustering image features, we calculate the average feature $\mu_i$ of unlabeled images from each domain $i$, representing the domain-specific feature for that domain. Also, we calculate the average feature of all unlabeled images $\mu_{\mathrm{avg}}$, representing the domain-invariant feature since it encompasses various domain distributions. Following Equation 5, the transition information $\widehat{t^i}$ of domain $i$ can be computed as $\widehat{t^i} = \mu_i - \mu_{\mathrm{avg}}$. By subtracting this domain transition vector, we suppress the preference for specific class names in the original text features. To ensure that the calibrated text features effectively eliminate biases towards a wide range of domains, we integrate the text features calibrated on each domain. Specifically, for the text feature $t_j$ of class $j$, we calibrate it as follows:

$$t'_j = \frac{1}{M} \sum_{i=1}^{M} \frac{t_j - \widehat{t^i}}{\left\| t_j - \widehat{t^i} \right\|_2}, \tag{6}$$

where $M$ is the number of clusters.

**Inference.** After calibration with IFC and TFC modules, we obtain the final prediction results. Specifically, this calibration modifies the prediction probability on test image $x$ as follows:

$$p(y_i|x) = \frac{\exp\left(\mathrm{sim}\left(f', t'_i\right)/\tau\right)}{\sum_{j=1}^{K} \exp\left(\mathrm{sim}\left(f', t'_j\right)/\tau\right)}. \tag{7}$$

## 5 Experiments

### 5.1 Experimental Setting

**Datasets.** Our UMFC is *training free*, which only calibrates the image and text features using Equation 4 and 6. To analyze model's generalization capability, we use two large-scale datasets for evaluation: 1) *DomainNet* [29] consists of 569,010 images with 345 categories from six domains: Clipart (C), Infograph (I), Painting (P), Quickdraw (Q), Real (R), Sketch (S). 2) *ImageNet Variants* composed of several datasets shifted from ImageNet, including ImageNet-A (IN-A) [13], ImageNet-R (IN-R) [12], and ImageNet-Sketch (IN-S) [36]. We form the class space for *ImageNet Variants* by taking the union of the class sets in IN-A and IN-R. To ensure the reliability of the evaluation results, we randomly sample the test data to construct a balanced test set where both domain and category distributions are uniform.

**Evaluation Paradigms.** Our approach is a universal feature calibration technique that can be applied across multiple scenarios. In this work, we explore three settings: 1) *Unsupervised Calibration* (UC) where the unlabeled training set is provided for computing the calibration vectors; 2) *Transductive Learning* (TL) where the entire unlabeled test set is provided at once, without providing any training data; 3) *Test-Time Adaptation* (TTA) where the model can be adapted to test samples shifted from training distribution. Different from TL, the test data usually arrives in batches in TTA setting. More details on the experimental setup, please refer to Appendix D.

### 5.2 Main Results on Unsupervised Calibration.

**Implementation Details.** We select CLIP [30] as our pre-trained vision-language model. We use CLIP with ViT-B/16 [6] as image encoder, and keep the original transformer as the text encoder. By default, a fixed prompt, "a photo of a [class]", is employed for all datasets. The images are resized to $224 \times 224$. The hyper-parameter $M$ (cluster number) is set to 6 for DomainNet. Remarkably, our method is training-free where both image encoder and text encoder remain frozen throughout the entire pipeline. All experiments are performed on a GeForce RTX 3090 Ti GPU.

**Baselines.** We compare our method with four groups of methods: (1) CLIP and its variants to show zero-shot predictions: CLIP [30] that uses a fixed prompt "a photo of a [class]"; CLIP-E [30] that uses an ensemble of prompt templates. (2) CLIP-D [30] that utilizes the domain information of test samples and designs domain-specific prompts. (3) Few-shot learning methods: CoOp [39] that

Table 1: Results on DomainNet under multi-domain Unsupervised Calibration. CLIP denotes zero-shot CLIP with a fixed text prompt template "a photo of a [class]", CLIP-E uses the ensemble prompt templates designed for Imagenet [39], CLIP-D uses the domain-specific templates, i.e., "a [domain] image of [class]". CoOp and CLIP-Adapter are trained on multi-domain labeled data, e.g., $6 \times 1 \times 345$ denotes the number of labeled data.

| | Method | C | I | P | Q | R | S | Avg |
|---|---|---|---|---|---|---|---|---|
| Unsupervised | CLIP [30] | 71.21 | 49.47 | 64.61 | 14.23 | 82.98 | 64.81 | 57.88 |
| | CLIP-E [30] | 73.16 | 54.17 | 67.02 | 15.86 | 84.30 | 67.49 | 60.33 |
| | CLIP-D [30] | 73.90 | 55.84 | 67.75 | 17.84 | 83.26 | 67.56 | 61.03 |
| | MUST [25] | 74.83 | 56.48 | 61.80 | 19.06 | 82.88 | 70.31 | 60.89 |
| | **UMFC** (ours) | 73.02 | 55.30 | 66.36 | 19.67 | 83.54 | 66.87 | 60.79 |
| | **UMFC + CLIP-E** (ours) | 73.84 | 56.59 | 67.39 | 20.03 | 84.33 | 67.90 | **61.68** |
| Few-Shot | CoOp ($6 \times 1 \times 345$) [39] | 72.73 | 53.95 | 66.80 | 19.58 | 82.53 | 67.27 | 60.48 |
| | CoOp ($6 \times 4 \times 345$) [39] | 74.7 | 54.96 | 68.29 | 22.14 | 82.94 | 69.48 | 62.09 |
| | CLIP-Adapter ($6 \times 1 \times 345$) [11] | 72.67 | 51.69 | 67.84 | 17.82 | 84.26 | 65.75 | 60.00 |
| | CLIP-Adapter ($6 \times 4 \times 345$) [11] | 74.35 | 53.79 | 69.94 | 19.71 | 85.26 | 66.90 | 61.66 |

performs prompt tuning using labeled data of downstream tasks; CLIP-Adapter [11] that trains an adapter using task-specific labeled data. (4) Unsupervised Fine-tuning method: MUST [25] that fine-tunes the model using unlabeled multi-domain data.

**Analysis.** As shown in Table 1, UMFC achieves superior performance over CLIP and competitive performance with few-shot methods, CoOp [39] and CLIP-Adapter [11]. We can observe that: **(1)** CLIP-D incorporates domain information into text prompts, such as "a clipart image of a [class]" for test samples from "clipart" domain. However, creating domain-specific templates is challenging due to unknown and potentially mixed domain sources of test samples. So, we only use CLIP-D as an oracle result. As shown in Table 1, our method can achieve competitive performance with CLIP-D without prior domain labels for test samples. **(2)** On domains where CLIP performs poorly, e.g., "quickdraw", our UMFC significantly outperforms others. Compared to CLIP, UMFC achieves about 5% performance gain on quickdraw domain. In addition, combined with more diverse prompts of CLIP-E, UMFC further improves performance. **(3)** MUST [25] uses abundant unlabeled data from 6 domains for fine-tuning. Our UMFC achieves better performance than MUST even without any additional training. **(4)** For CoOp [39] and CLIP-Adapter [11], we fine-tune them using labeled samples from multiple domains. Specifically, we provide $k$ labeled samples per class for each domain, resulting in a total of $(6 \times k \times 345)$ labeled samples available for training. When the number of labeled samples is $6 \times 345$, our method outperforms few-shot fine-tuning methods. While the performance is higher for few-shot methods with a larger number of labeled samples (i.e., $24 \times 345$), it is essential to highlight the challenges in obtaining some labeled data for each class and each domain in real-world scenarios. In contrast, our method does not require selecting class-balanced and domain-balanced labeled samples and incurs no additional training overhead.

Table 2 provides $8 \times 345$ labeled samples from a single domain for CoOp fine-tuning and an equal number of unlabeled samples for UMFC calibration. While CoOp trained on a single domain can improve performance within that domain, its performance declines on other domains. This decline becomes particularly notable when a significant distribution gap between the training and test domains exists, e.g., CoOp (Q), leading to a large decrease in average performance across multiple domains. In contrast, our method achieves consistent performance improvements on both training and unseen domains with the same amount of training data. Additionally, unlike CoOp, UMFC does not require labeled data or any parameter fine-tuning.

### 5.3 Main Results on Transductive Learning

In this part, we compare our UMFC with three groups of methods: (1) Zero-Shot CLIP models. (2) Domain Generalization (DG) method: MIRO [2] that trains CLIP on available data to learn great generalization capability. (3) Data-Free DG method: PromptStyler [5] that learns to simulate diverse distributions. The hyper-parameter $M$ (cluster number) is set to 3 for ImageNet-Variants. The compared results are shown in Table 3 and Table 4. As seen, our UMFC can achieve the best average performance on DomainNet and ImageNet-Variants benchmarks, which validates the effectiveness of our approach in transductive learning.

Table 2: Results on DomainNet under single-domain Unsupervised Calibration. $8 \times 345$ samples (each class has 8 samples) from a single domain are provided. CoOp (C/Q/I) and UMFC (C/Q/I) denote training samples for CoOp and UMFC from the "Clipart"/"Quickdraw"/"Infograph" domains, respectively.

| Method | C | I | P | Q | R | S | Avg |
|---|---|---|---|---|---|---|---|
| CLIP [30] | 71.21 | 49.47 | 64.61 | 14.23 | 82.98 | 64.81 | 57.88 |
| CoOp (C) [39] | 74.55 | 42.66 | 55.94 | 13.82 | 75.00 | 58.73 | 53.45 |
| **UMFC** (C) | 73.27 | 52.96 | 65.27 | 16.94 | 83.60 | 67.04 | **59.85** |
| CoOp (Q) [39] | 43.97 | 25.5 | 32.63 | 29.07 | 48.44 | 38.74 | 36.39 |
| **UMFC** (Q) | 72.17 | 49.65 | 63.85 | 17.47 | 82.84 | 66.36 | **58.72** |
| CoOp (I) [39] | 60.19 | 54.28 | 50.81 | 11.19 | 70.73 | 54.27 | 50.24 |
| **UMFC** (I) | 72.54 | 55.21 | 64.48 | 16.30 | 83.31 | 66.51 | **59.73** |

Table 3: Comparison Results on DomainNet under Transductive Learning.

| Method | C | I | P | Q | R | S | Avg |
|---|---|---|---|---|---|---|---|
| CLIP [30] | 71.21 | 49.47 | 64.61 | 14.23 | 82.98 | 64.81 | 57.88 |
| CLIP-E [30] | 73.16 | 54.17 | 67.02 | 15.86 | 84.30 | 67.49 | 60.33 |
| CLIP-D [30] | 73.90 | 55.84 | 67.75 | 17.84 | 83.26 | 67.56 | 61.03 |
| MIRO [2] | - | - | - | - | - | - | 54.00 |
| PromptStyler [5] | 73.10 | 50.90 | 69.20 | 13.30 | 85.40 | 65.30 | 59.40 |
| **UMFC** | 73.01 | 55.44 | 66.89 | 20.14 | 83.66 | 67.51 | **61.11** |

Table 4: Comparison Results on ImageNet-Variants under Transductive Learning.

| Method | IN-A | IN-R | IN-S | Avg |
|---|---|---|---|---|
| CLIP [30] | 42.13 | 66.95 | 74.58 | 61.22 |
| CLIP-E [30] | 45.42 | 71.10 | 77.08 | 64.53 |
| **UMFC** | 43.42 | 68.86 | 77.24 | 63.17 |
| **UMFC + CLIP-E** | 44.77 | 72.19 | 78.62 | **65.19** |

Table 5: Comparison Results on ImageNet-Variants under Test-Time Adaptation.

| Method | IN-A | IN-R | IN-S | Avg |
|---|---|---|---|---|
| CLIP | 42.13 | 66.95 | 74.58 | 61.22 |
| TPT [32] | 47.16 | 59.95 | 67.49 | 58.20 |
| **UMFC-Memory** | 42.76 | 67.03 | 75.15 | 61.65 |
| **UMFC-EMA** | 42.21 | 67.82 | 76.26 | **62.10** |

## 5.4 Main Results on Test-Time Adaptation

**Implementation Details.** In TTA setting, where we cannot access all data simultaneously, we adopt an incremental clustering approach. By default, the batch size is set to 100. Initially, we apply K-Means clustering algorithm to the first batch of data. For subsequent batches, we use the prototype classification, with cluster centers serving as prototypes, to assign samples in that batch to different clusters. Then, the cluster centers and calibration statics $\{\mu_i\}_{i=1}^{M}$ are updated accordingly. We have developed two strategies for updating the calibration statics: **UMFC-Memory** that stores the feature information of each batch to calculate the statistical information; **UMFC-EMA** that uses Exponential Moving Average (EMA) to update statistical information.

Table 6: Comparison Results on DomainNet under Test-Time Adaptation. UMFC-Memory and UMFC-EMA represent different ways to update the statics vectors for calibration.

| Method | C | I | P | Q | R | S | Avg |
|---|---|---|---|---|---|---|---|
| CLIP [30] | 71.21 | 49.47 | 64.61 | 14.23 | 82.98 | 64.81 | 57.88 |
| TPT [32] | 73.23 | 52.63 | 68.00 | 12.79 | 84.39 | 66.68 | 59.62 |
| **UMFC-Memory** | 72.82 | 55.12 | 66.82 | 19.92 | 83.62 | 66.82 | **60.85** |
| **UMFC-EMA** | 72.99 | 54.94 | 66.64 | 18.58 | 83.54 | 66.75 | 60.57 |

**Analysis.** We mainly compare our UMFC with TPT [32] in test-time adaptation, where TPT fine-tunes the prompts by minimizing the entropy of the predictions. The comparison results are shown in Table 6 and Table 5. We can observe that: **(1)** As shown in Table 6, UMFC achieves performance gains across all domains, with particularly noticeable gains in those where CLIP performs poorly. For example, UMFC substantially improves upon CLIP on "quickdraw" and "infograph" domains, with

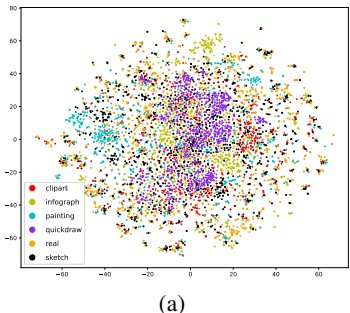
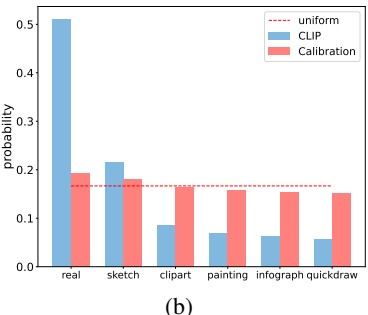

| (a) | (b) |

Figure 2: On DomainNet dataset, we visualize (a) The image features extracted by UMFC image encoder across different domains. (b) The classification probabilities of CLIP's text features on different domains.

Table 7: Ablation study on the effects of TFC and IFC under Transductive Learning.

| Method | C | I | P | Q | R | S | Avg |
|---|---|---|---|---|---|---|---|
| CLIP [30] | 71.21 | 49.47 | 64.61 | 14.23 | 82.98 | 64.81 | 57.88 |
| IFC | 72.98 | 55.07 | 66.65 | 19.87 | 83.54 | 66.97 | 60.85 |
| TFC | 71.44 | 49.99 | 65.59 | 13.9 | 83.25 | 64.68 | 58.14 |
| **UMFC** | 73.01 | 55.44 | 66.89 | 20.14 | 83.66 | 67.51 | **61.11** |

an accuracy gain of $5.6\%$. Table 5 shows that the improvement on ImageNet-A is less significant than on ImageNet-R and ImageNet-S. The reason may be attributed to the absence of distinct domain styles in ImageNet-A, limiting the effectiveness of our calibration method that relies on domain information. **(2)** UMFC-Memory and UMFC-EMA with different statistical update strategies exhibit similar performance. However, UMFC-EMA, which updates features based on the most recent data, notably reduces storage requirements. Moreover, UMFC exhibits significantly higher computational efficiency without training, whereas TPT requires executing optimization steps on $64$ different augmented views of each test image. Thus, our method is more suitable for rapid deployment.

### 5.5 Ablation Study

**The effectiveness of IFC.** We firstly evaluate the effectiveness of IFC. As shown in Table 7, IFC individually contributes to performance gains of CLIP, about $3\%$ average gains. Furthermore, Figure 1(b) and Figure 2(a) visualize the image features extracted by CLIP with/without IFC respectively. As shown in Figure 1(b), the vanilla CLIP maps different-domain images to different clusters in the feature space. Conversely, Figure 2(a) shows that IFC leads to the merge of image features from different domains, validating its effectiveness of reducing domain-specific information.

**The effectiveness of TFC.** As shown in Table 7, TFC also individually contributes to performance gains of CLIP. To assess the effectiveness of calibrated text features in eliminating domain bias, we construct a domain classifier. Specifically, we utilize the text features generate by CLIP, with domain prompts "[domain]", as the domain classifer. Then, we use this domain classifier to perform domain classification for the text features before and after calibration (*i.e.*, computing the cosine similarity between domain classifier and text features). As shown in Figure 2(b), the original text features exhibit a long-tail phenomenon, with a higher probability of being classified into the "real" domain. This suggests that the original text features are biased towards the "real" domain, overlooking their generalization ability to other domains. However, after calibration with TFC, the text features exhibit a near-uniform distribution across domains, indicating that the calibrated text features have largely eliminated domain bias. Consequently, our method enhances performance in other domains without compromising the model's performance in its strongest domain. As shown in Table 7, the combination of IFC and TFC (*i.e.*, UMFC) can further bring performance gains, which validates the complementary of image-level calibration and text-level calibration for CLIP.

**The impact of cluster number $M$.** Our method involves clustering the unlabeled data to determine their respective clusters. We evaluate our method with respect to the number of clusters $M$. As shown

Table 8: The impact of cluster number $M$ on DomainNet under Transductive Learning.

| Method | C | I | P | Q | R | S | Avg |
|--------|-----|-----|-----|-----|-----|-----|-----|
| CLIP [30] | 71.21 | 49.47 | 64.61 | 14.23 | 82.98 | 64.81 | 57.88 |
| UMFC (M=3) | 72.53 | 54.60 | 66.31 | 20.32 | 83.35 | 66.86 | 60.66 |
| UMFC (M=4) | 73.55 | 56.36 | 67.19 | 20.62 | 84.13 | 67.69 | 61.59 |
| UMFC (M=6) | 73.01 | 55.44 | 66.89 | 20.14 | 83.66 | 67.51 | 61.11 |
| UMFC (M=8) | 73.50 | 56.58 | 67.53 | 20.64 | 84.06 | 67.92 | 61.71 |
| UMFC (M=10) | 73.63 | 56.87 | 67.81 | 20.23 | 84.20 | 67.87 | **61.77** |

Table 9: The impact of batch size under Test-Time Adaptation.

| Method | C | I | P | Q | R | S | Avg |
|--------|-----|-----|-----|-----|-----|-----|-----|
| CLIP [30] | 71.21 | 49.47 | 64.61 | 14.23 | 82.98 | 64.81 | 57.88 |
| UMFC (bs=1) | 72.64 | 53.74 | 66.39 | 18.25 | 83.34 | 66.90 | 60.21 |
| UMFC (bs=10) | 72.64 | 54.52 | 66.51 | 18.53 | 83.35 | 67.06 | 60.44 |
| UMFC (bs=16) | 72.70 | 54.80 | 66.91 | 19.11 | 83.78 | 66.69 | 60.66 |
| UMFC (bs=32) | 73.02 | 55.02 | 66.73 | 19.17 | 83.66 | 66.97 | 60.76 |
| UMFC (bs=64) | 73.23 | 55.04 | 66.72 | 19.15 | 83.78 | 66.84 | 60.79 |
| UMFC (bs=100) | 72.82 | 55.12 | 66.82 | 19.92 | 83.62 | 66.82 | **60.85** |

in Table 8, our UMFC consistently outperforms vanilla CLIP, even when the number of clusters $M$ does not match the number of domains (6 for DomainNet). More importantly, our method is not sensitive to changes in $M$. Refer to the Appendix E for more analysis.

**The impact of batch size in TTA setting.** We present results across various batch sizes during test-time adaptation to confirm that UMFC is robust to batch size variations. When the sample count is initially lower than the number of clusters $M$, K-Means clustering cannot be directly applied. To address this, we used the first $M$ samples as the initial cluster centers and then proceeded with the same test-time adaptation. As shown in Table 9, even in the extreme case of a batch size of 1, our method still demonstrates consistent improvement.

## 6 Conclusion

In this work, we point out that the model biases hinder the transfer ability of pre-trained vision-language models. We develop UMFC, a simple unsupervised calibration method that mitigates model biases in both visual encoder and text encoder through a training-free manner. We demonstrate the effectiveness of our method across multiple settings, including unsupervised calibration, transductive learning, and test-time adaptation. Without the need for any annotations and training, UMFC improves the zero-shot generalization ability of CLIP.

**Limitations and Broader Impacts.** Our method requires unlabeled samples from the target domain to establish domain-level calibration vectors. Although our method is much cheaper and more accessible than fine-tuning or few-shot methods [39, 38, 32], the requirement of unlabeled data can still be a limitation for some specific scenarios.

## Acknowledgments

This work is partially supported by National Science and Technology Major Project No. 2021ZD0111901, National Natural Science Foundation of China (NSFC): 62376259 and 62306301, National Postdoctoral Program for Innovative Talents under Grant BX20220310.

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

## A  The Prevalence of the Observed Model Bias

Our method is motivated by observed biases in CLIP. Therefore, it is valuable to investigate whether these observations hold when the pre-training distribution and model architecture are altered.

We further verify this on OpenCLIP[4]. OpenCLIP investigates scaling laws for contrastive language-image pre-training (CLIP) with the public LAION dataset and the open-source OpenCLIP repository. The large-scale experiments involve models trained on up to two billion image-text pairs (LAION-2B) and identify power law scaling for multiple downstream tasks including zero-shot classification, retrieval, linear probing, and end-to-end fine-tuning. We used OpenCLIP models trained on different datasets to obtain the corresponding image features, and visualized them using t-SNE. As shown in Figure 3, even when the training data changes, images from different domains still cluster together, demonstrating that our observation is universally applicable.

Additionally, we validated the effectiveness of our method on the OpenCLIP series models in Table 10. It is evident that the zero-shot capabilities of larger-scale models have significantly improved compared to CLIP, and our method still provides further performance gains. This demonstrates that our method is universally applicable to Vision-Language models with different architectures and training data.

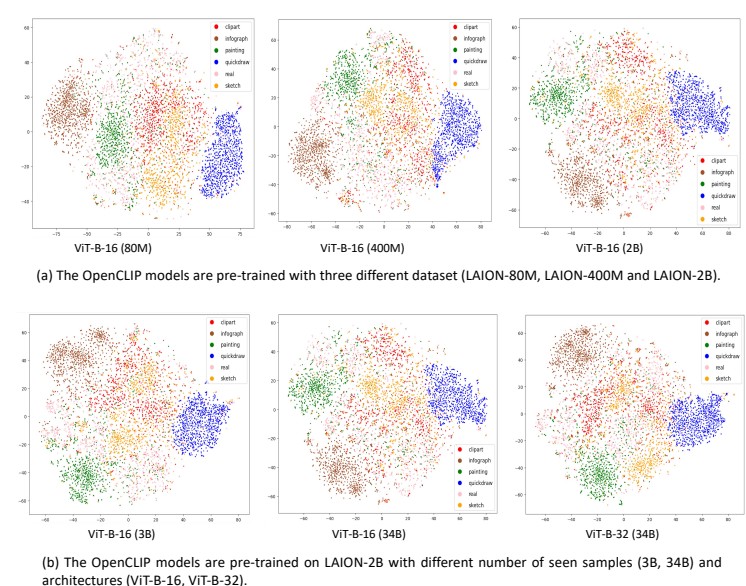

(a) The OpenCLIP models are pre-trained with three different dataset (LAION-80M, LAION-400M and LAION-2B).

(b) The OpenCLIP models are pre-trained on LAION-2B with different number of seen samples (3B, 34B) and architectures (ViT-B-16, ViT-B-32).

Figure 3: Visualization of Image Features based on OpenCLIP series.

Table 10: Comparision Results on DominaNet using OpenCLIP.

| Arch | C | I | P | Q | R | S | Avg |
|---|---|---|---|---|---|---|---|
| ViT-B-16 | 78.66 | 56.16 | 71.09 | 15.25 | 86.93 | 72.26 | 63.39 |
| + UMFC | 78.93 | 58.29 | 71.78 | 21.56 | 86.36 | 73.29 | **65.04** |
| ViT-B-32 | 76.98 | 52.00 | 68.77 | 15.63 | 85.63 | 70.54 | 61.59 |
| + UMFC | 76.99 | 53.08 | 68.70 | 22.61 | 85.17 | 71.27 | **62.97** |
| ViT-H-14 | 83.72 | 63.21 | 76.05 | 18.20 | 89.49 | 79.18 | 68.31 |
| + UMFC | 83.74 | 63.89 | 76.41 | 23.73 | 89.33 | 79.49 | **69.43** |

## B  Details of Text Feature Calibration Module

Since the domain labels are unavailable, we attempt to extract domain information from images in downstream tasks, and then transfer this domain information to the text embedding space to eliminate

the text bias. Inspired by Observation 4.1, we can estimate the domain transition direction in text embedding space by using images from different domains, as shown in Figure 4.

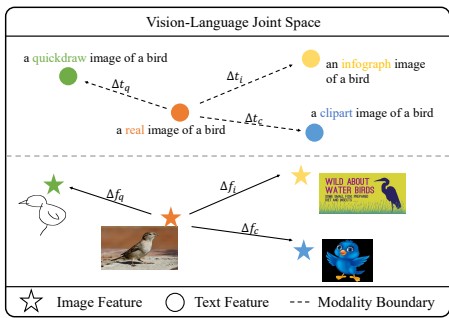

Figure 4: The domain transition direction between texts is similar to that between images.

# C  Algorithm

Algorithm 1 summarizes the proposed UMFC method under Test-Time Adaptation (TTA) setting.

Algorithm 2 summarizes the proposed UMFC method under Unsupervised Calibration (UC) / Transductive Learning (TL).

---

**Algorithm 1** Algorithm UMFC under TTA

---

1: **Input**: test data $\{u_{b,i}\}_{b=1}^{B}{}_{i=1}^{N}$, batch size $B$, total number of batches $N$, total number of clusters $M$, CLIP's image encoder $F(\cdot)$ and text encoder $T(p;\cdot)$, $p$ is the text template, class names $\{y_i\}_{i=1}^{C}$.

2: $t \leftarrow T(p; \{y_i\}_{i=1}^{C})$      ▷ Get original text features

3: Initialize centroids of $M$ cluster $\{c_j\}_{j=1}^{M}$ as empty set

4: **for** $i \leftarrow 1$ **to** $N$ **do**

5:    $\{f_{b,i}\}_{b=1}^{B} \leftarrow F(\{u_{b,i}\}_{b=1}^{B})$      ▷ Get original image features of $i$-th batch

6:    **if** $\{c_j\}_{j=1}^{M}$ is $\emptyset$ **then**

7:       $\{c_j\}_{j=1}^{M} \leftarrow$ K-Means$(\{f_{b,i}\}_{b=1}^{B})$      ▷ Using K-Means to acquire $M$ centroids

8:    **end if**

9:    **for** $b \leftarrow 1$ **to** $B$ **do**

10:       $l_{b,i} \leftarrow \arg\min_{m} \|f_{b,i} - c_m\|_2$      ▷ Assign sample to the nearest clusters

11:    **end for**

12:    $c_k' \leftarrow \frac{1}{B}\sum_{b=1}^{B} \mathbb{I}(l_{b,i} = m) \cdot f_{b,i}$      ▷ Calculate centroids of current batch

13:    $c_k \leftarrow c_k + \eta \cdot c_k'$,      ▷ Update prototypes, where $\eta$ is the update weight

14:    $\mu_{avg} \leftarrow \frac{1}{M}\sum_{m=1}^{M} c_m'$      ▷ update the average image feature

15:    $\hat{t}^m \leftarrow \mathbb{I}(l = m)(c_m' - \mu_{avg})$,      ▷ Update calibration statics

16:    $t_b' \leftarrow \frac{1}{M}\sum_{m=1}^{M} \frac{t - \hat{t}^m}{\|t - \hat{t}^m\|}$,

17:    **for** $b \leftarrow 1$ **to** $B$ **do**

18:       $f_{b,i}' \leftarrow \frac{f_{b,i} - c_l'}{\|f_{b,i} - c_l'\|}$,      ▷ Calibrate $f$ and $t$ with IFC and TFC

19:    **end for**

20: **end for**

21: **return** $\{f_{b,i}', t_b'\}_{b=1}^{B}{}_{i=1}^{N}$      ▷ Calibrated image and text features

---

**Algorithm 2** Algorithm UMFC under UC / TL
___

1: **Input**: unlabeled data $\{u_b\}_{i=1}^N$, total number of samples $N$, total number of clusters $M$, total number of batches $B$, CLIP's image encoder $F(\cdot)$ and text encoder $T(p;\cdot)$, $p$ is the text template, class names $\{y_i\}_{i=1}^C$.

2: $t \leftarrow T(p; \{y_i\}_{i=1}^C)$           $\triangleright$ Get original text features

3: $\{f_b\}_{b=1}^N \leftarrow F(\{u_b\}_{i=1}^N)$           $\triangleright$ Get original image features of unlabeled data

4: $\mu_{avg} \leftarrow \frac{1}{N} \sum_{b=1}^N f_b$           $\triangleright$ Get the average of $\{f_b\}_{b=1}^N$

5: $\{c_j\}_{j=1}^M \leftarrow$ K-Means$(\{f_b\}_{b=1}^N)$     $\triangleright$ Using K-Means to acquire $M$ clusters and their centroids

6: **for** $b \leftarrow 1$ **to** $N$ **do**

7:     $l \leftarrow \underset{m}{\arg\min} \|f_b - c_m\|_2$           $\triangleright$ Assign sample to the nearest clusters

8:     $\hat{t}_k \leftarrow \mathbb{I}(l = m)(c_m - \mu_{avg})$,           $\triangleright$ Update calibration statics

9:     $f'_b \leftarrow \frac{f_b - c_l}{\|f_b - c_l\|}, t'_b \leftarrow \frac{1}{M} \sum_{m=1}^M \frac{t - \hat{t}^m}{\|t - \hat{t}^m\|}$,       $\triangleright$ Calibrate $f$ and $t$ with IFC and TFC

10: **end for**

11: **return** $\{f'_b\}_{b=1}^N, t'$           $\triangleright$ Calibrated image and text features
___

# D    Experimental Settings

Our work can be deployed across various scenarios, including Unsupervised Calibration (UC), Test-Time Adaptation (TTA), and Transductive Learning (TL).

**Unsupervised Calibration.** In the UC scenario, we provide an unsupervised training set and use K-Means to assign cluster labels to the training samples, while also saving the corresponding cluster prototypes. Then, we create an unlabeled training set from a mixed domain by sampling 16 instances from each class across 6 domains in DomainNet. UMFC derives image bias and text bias for different clusters based on the cluster labels. During the testing phase, we first predict the cluster labels of the test samples using the cluster prototypes obtained during training, and then calibrate the predictions using the bias information derived from UMFC.

**Test-Time Adaptation.** In the TTA scenario, no training data is provided. Test data from mixed domains arrive in batches, with a batch size set to 100. For the first batch, we perform initial clustering using K-Means. For subsequent batches, we assign cluster labels to the samples based on cluster prototypes and continuously update these prototypes. Once the cluster labels are obtained, UMFC calculates the bias information for the current batch, updates the bias information for each cluster based on the new labels, and then calibrates the data for the current batch.

**Transductive Learning.** In the TL scenario, we provide the entire test set. Similar to UMFC, we gather statistical information and calibrate the predictions for the test data. TL can be viewed as an extreme case of TTA, where the entire test set is treated as a single batch.

# E    Experimental Analysis

**The impact of cluster number $M$.** We evaluate our method with respect to the number of clusters $M$ and demonstrate that our method is not sensitive to the choice of this hyperparameter. As shown in Table 11, For instance, setting $M$ from 2 to 6 all leads to improvements on ImageNet-Variants.

Table 11: The impact of cluster number $M$ on ImageNet-Variants under Transductive Learning.

| Method | IN-A | IN-R | IN-S | Avg |
|---|---|---|---|---|
| CLIP [30] | 42.13 | 66.95 | 74.58 | 61.22 |
| UMFC (M=2) | 45.35 | 71.71 | 77.37 | 64.81 |
| UMFC (M=3) | 44.77 | 72.19 | 78.62 | **65.19** |
| UMFC (M=6) | 45.29 | 71.33 | 77.59 | 64.74 |

**Computation Cost.** For analyzing computational cost, we report the training time, inference time and memory of UMFC and other comparison methods under different scenarios. In **Unsupervised**

**Calibration** scenario, the entire unlabeled training set is provided for training. Corresponding computation cost comparisons are shown in Table 12. Firstly, UMFC incurs minimal training and inference overhead compared to CLIP. This is because UMFC only requires a single forward pass to extract features and then calculate statistics for feature calibration. Secondly, when compared to few-shot fine-tuning methods (CoOp) and unsupervised fine-tuning methods (MUST), UMFC also demonstrates lower consumption of computational resources and time. In **Test-time Adaptation** scenario, no training data is provided and the test data arrives in batches. Corresponding computation cost comparisons are shown in Table 13. UMFC requires less memory than TPT and shows greater computational efficiency. Specifically, UMFC takes only 296 seconds, whereas TPT requires nearly 197 minutes. This is because TPT requires fine-tuning the text prompt for each test sample and augmenting each test sample 64 times to ensure the reliability of the fine-tuning results, which significantly slows down TPT's inference speed.

Table 12: Computation Cost under Transductive Learning.

| Method | Training Time | Inference Time | Epoch | Memory |
|---|---|---|---|---|
| CLIP [30] | - | 86 seconds | - | 1797MiB |
| MUST [25] | 10 hours (2 GPUs) | 92 seconds | 30 | 25944MiB |
| CoOp (6*1 shot) [39] | 32 minutes | 83 seconds | 50 | 7007MiB |
| CoOp (6*4 shots) [39] | 160 minutes | 83 seconds | 100 | 7007MiB |
| UMFC | 57.3 seconds | 86 seconds | - | 1887MiB |

Table 13: Computation Cost under Test-Time Adaptation.

| Method | Inference Time | Memory |
|---|---|---|
| TPT | 197 minutes | 6872MiB |
| UMFC | 296 seconds | 1790MiB |

