# OpenReview forum: "UMFC: Unsupervised Multi-Domain Feature Calibration for Vision-Language Models"
_NeurIPS.cc/2024/Conference — NeurIPS 2024 poster_

### Official Review · Reviewer_hYxJ · 2024-07-10

**Soundness:** 3
**Presentation:** 3
**Contribution:** 3
**Rating:** 6
**Confidence:** 4

**Summary:**

To boost the transferability of CLIP across downstream domains, the paper proposed Multi-Domain Feature Calibration (UMFC). By mitigating CLIP biases in both visual and text encoders, UMFC significantly improves classification performance over existing methods across 3 downstream tasks.

**Strengths:**

1. The observation of domain-dependent performance variability in CLIP is interesting. The observation highlights that the accuracy of CLIP can vary across different domains for the same classes.
2. The paper is well-motivated. The author starts with the bias phenomenon empirically and further presents the motivation from a probabilistic view.
3. The experiment is comprehensive. The efficacy of UMFC is verified on three downstream tasks and extensive experiments demonstrate the consistent gains.

**Weaknesses:**

1. The proposed methods rely on multi-domain data for calibration, which slightly weakens its application scenarios.
2. The comparison in TTA is not enough. As the paper use prototype for calibration, the proposed method be compared with other state-of-the-art prototype-based methods such as T3A [1].
3. The ablation of the number of clusters \textit{M} is insufficient. Since we do not know the number of domains, what would the results be if we set \textit{M} higher than the actual number of domains?

[1] Iwasawa Y, Matsuo Y. Test-time classifier adjustment module for model-agnostic domain generalization. In NeurIPS, 2021.

**Questions:**

1. The author set the batch size of TTA to 100. Is the proposed method sensitive to the batch size?

---

> ### Author Rebuttal · Authors · 2024-08-07
>
> **W1: The proposed methods rely on multi-domain data for calibration, which slightly weakens its application scenarios.**
>
> (1) In fact, our method is not limited to multi-domain scenarios; it is also applicable to single-domain scenarios. In single-domain case, we employ the same calibration process used in UMFC. This involves gathering statistical information from the clustered training set and utilizing this information to calibrate the features.
>
> (2) Following the unsupervised calibration setting in Section 5.2, we conduct experiments on DomainNet, where both the unlabeled training data and test data originate from the same domain. As shown in the Table below, UMFC outperforms vanilla CLIP even in single-domain scenarios, and its  performance is only marginally affected by the number of clusters (M).
>
> |      | C     | I     | P     | Q     | R     | S     |
> | ---- | ----- | ----- | ----- | ----- | ----- | ----- |
> | CLIP | 71.21 | 49.47 | 64.61 | 14.23 | 82.98 | 64.81 |
> | UMFC (M=3)    | 73.90   | 56.69  | 68.12   | 20.43   | 84.70   | 68.04   |
> | UMFC (M=1) | 73.85 | 56.7 | 68.02 | 20.31 | 84.67 | 68.17 |
>
>
> **W2: The comparison in TTA is not enough. As the paper use prototype for calibration, the proposed method be compared with other state-of-the-art prototype-based methods such as T3A [1].**
>
> Thanks for pointing out this. In the following, we compare T3A and our UMFC under test-time adaptation setting. To ensure a fair comparison, we integrated T3A into CLIP framework, using CLIP model instead of a pre-trained source domain model.  Additionally, we set the hyperparameter *N* in T3A to 1 (the optimal value in our experiment), which determines the number of supports to restore. We set the batch size to 100 the same as UMFC. The results are shown in the below table.
>
> As shown in the below table, T3A's performance improved only in the Infograph (I) domain but declined in other domains, particularly in the Quickdraw (Q) domain. We attribute this to  domain bias inherent in CLIP, which is exacerbated by T3A. In contrast, our UMFC leverages the characteristics of CLIP across different domains for calibration, achieving better performance across all domains.
>
>
> |      | C     | I     | P     | Q     | R     | S     | Avg   |
> | ---- | ----- | ----- | ----- | ----- | ----- | ----- | ----- |
> | CLIP | 71.21 | 49.47 | 64.61 | 14.23 | 82.98 | 64.81 | 57.88 |
> | T3A  | 65.56 | 51.55 | 62.02 | 1.80  | 80.49 | 59.53 | 53.49 |
> | UMFC | **72.82** | **55.12** | **66.82** | **19.92** | **83.62** | **66.82** | **60.85**|
>
>
> **W3: The ablation of the number of clusters M is insufficient. Since we do not know the number of domains, what would the results be if we set M higher than the actual number of domains?**
>
> Thanks for pointing out this. We conduct ablation studies on cluster number *M* on both DomainNet and ImageNet-Variants (see the table in **Global Response 2**). The results indicate that our method is not sensitive to the value of M, as it still delivers significant improvements even when *M* does not match the actual number of domains (8/10 for DomainNet and 6 for ImageNet-Variants).
>
>
> **Q1: The author set the batch size of TTA to 100. Is the proposed method sensitive to the batch size?**
>
> In the table below, we present the results of UMFC with different batch sizes in test-time adaptation scenario. The results validate that our proposed method is not sensitive to the batch size.
>
>
> |               | C     | I     | P     | Q     | R     | S     | Avg   |
> | ------------- | ----- | ----- | ----- | ----- | ----- | ----- | ----- |
> | CLIP          | 71.21 | 49.47 | 64.61 | 14.23 | 82.98 | 64.81 | 57.88 |
> | UMFC (bs=10)  | 72.64 | 54.52 | 66.51 | 18.53 | 83.35 | 67.06 | 60.44 |
> | UMFC (bs=16)  | 72.70 | 54.80 | 66.91 | 19.11 | 83.78 | 66.69 | 60.66 |
> | UMFC (bs=32)  | 73.02 | 55.02 | 66.73 | 19.17 | 83.66 | 66.97 | 60.76 |
> | UMFC (bs=64)  | 73.23 | 55.04 | 66.72 | 19.15 | 83.78 | 66.84 | 60.79 |
> | UMFC (bs=100) | 72.82 | 55.12 | 66.82 | 19.92 | 83.62 | 66.82 | **60.85** |

---

> > ### Comment · Reviewer_hYxJ · 2024-08-11
> >
> > Thanks for the response.  Actually, I want to see the performance in some extreme but realistic cases (e.g., BS=1).
> >
> > Overall, I will keep my positive score.
> >
> > Best

---

> > > ### Author Response · Authors · 2024-08-12
> > > **Response to the comment**
> > >
> > > **Actually, I want to see the performance in some extreme but realistic cases (e.g., BS=1).**
> > >
> > > Thank you for your response.
> > > We have added experimental results for the scenario where the batch size is 1.
> > > Initially, when the number of samples is less than the number of clusters \(M\), K-Means clustering cannot be applied.
> > > To address this, we used the first \(M\) samples as the initial cluster centers and then continued with the same Test-Time Adaptation.
> > > As shown in the table below, even in the extreme case of a batch size of 1, our method still demonstrates consistent improvement.
> > >
> > >
> > > |       | C     | I     | P     | Q     | R     | S     | Avg   |
> > > | ----- | ----- | ----- | ----- | ----- | ----- | ----- | ----- |
> > > | CLIP | 71.21 | 49.47 | 64.61 | 14.23 | 82.98 | 64.81 | 57.88 |
> > > | UMFC (bs=1) | 72.64 | 53.74 | 66.39 | 18.25 | 83.34 | 66.90 | 60.21 |

---

### Official Review · Reviewer_jj75 · 2024-07-12

**Soundness:** 3
**Presentation:** 3
**Contribution:** 2
**Rating:** 4
**Confidence:** 3

**Summary:**

This paper proposes a training-free method "Unsupervised Multi-domain Feature Calibration ( UMFC)" to mitigate the model bias problem presented in CLIP models. The paper first observes the bias problem from the visual encoder and the text encoder perspective. Then it mitigates the problem by subtracting the bias term in the visual and text features obtained from the domain data. The experimental results show improvements over existing approaches like CoOp or CLIP-Adapter.

**Strengths:**

1. The observation is clear. The paper provides a clear observation regarding the inherent biases in CLIP's visual and text encoders, which lead to biased prediction.
2. The approach is simple and straightforward. The proposed UMFC effectively leverages unlabeled data to mitigate domain biases without involving fine-tuning or extra optimization.
3. The presentation of the method is clear and its evaluation is clear and easy to follow.

**Weaknesses:**

1. I think the improvements seem minor. In table 1, compared to the really easy approach CLIP-D, UMFC fails to improve it and UMFC + CLIP-E outperforms it by only 0.65% points. What about CLIP-D + CLIP-E? I think you should use this one to agast UMFC + CLIP-E.
2. In table 2, why did CoOp perform worse than the original CLIP? I believe it's the receipt problem, and I think you should at least perform some hyper-parameter choice to get a reasonable baseline, since finetuning on those in-domain data can easily get performance improvements.
3. What's the extra computation cost involved in this approach and how it compares to other methods? Since the proposed approach uses test-time adaptation, I believe the authors should tell us the computation cost.

**Questions:**

Please see the weakness part.

---

> ### Author Rebuttal · Authors · 2024-08-07
>
> **W1: I think the improvements seem minor. In table 1, compared to the really easy approach CLIP-D, UMFC fails to improve it and UMFC + CLIP-E outperforms it by only 0.65% points. What about CLIP-D + CLIP-E? I think you should use this one to agast UMFC + CLIP-E.**
>
> There are some misunderstandings here, and I would like to clarify them in the following.
>
> - **Minor improvement compared to CLIP-D.** In CLIP-D, we design customized prompts that incorporates corresponding domain label and name for each test sample. CLIP-D shows better performance than vallina CLIP, demonstrating the benefit of integrating domain information in multi-domain scenarios. Notably, our comparison between UMFC and CLIP-D is not meant to imply that UMFC outperforms CLIP-D, as CLIP-D utilizes domain labels and names for all test samples, whereas UMFC operates without such explicit supervision. Rather, the purpose of this comparison is to demonstrate that UMFC can extract domain information from data **without supervision**, highlighting it greater versatility in various situations.
>
> - **Disadvantages of CLIP-D.** The CLIP-D model, while simple and straightforward, has several disadvantages: 1. it requires domain labels and names for all test samples, which is impractical in many real-world scenarios. 2. it is sensitive to the choice of domain names; Our findings show that CLIP-D's performance degrades when synonyms are used instead of the original domain names. Detailed results can be seen the below table.
>
>   |                   | C     | I     | P     | Q     | R     | S     | Avg   |
>   | ----------------- | ----- | ----- | ----- | ----- | ----- | ----- | ----- |
>   | CLIP-D            | 73.90 | 55.84 | 67.75 | 17.84 | 83.26 | 67.56 | 61.03 |
>   | CLIP-D (Synonyms) | 72.11 | 54.83 | 66.09 | 17.73 | 83.34 | 65.74 | 59.97 |
>
> - **More comparison baseline.** We build  CLIP-DE by replacing the single domain-specific prompt in CLIP-D with an ensemble of prompt template incorporating domain names. The results are shown in below table. As shown, CLIP-DE performs only marginally better than CLIP-E, and slightly worse than CLIP-D on DomainNet. This suggests that CLIP-D may require a more refined prompt design and ensemble strategy to improve performance in multi-domain scenarios.
>
>   |         | C     | I     | P     | Q     | R     | S     | Avg   |
>   | ------- | ----- | ----- | ----- | ----- | ----- | ----- | ----- |
>   | CLIP-E  | 73.16 | 54.17 | 67.02 | 15.86 | 84.30 | 67.49 | 60.33 |
>   | CLIP-D  | 73.90 | 55.84 | 67.75 | 17.84 | 83.26 | 67.56 | 61.03 |
>   | CLIP-DE | 73.62 | 54.34 | 67.64 | 16.34 | 84.49 | 67.45 | 60.65 |
>
>
>
> **W2: In table 2, why did CoOp perform worse than the original CLIP? I believe it's the receipt problem, and I think you should at least perform some hyper-parameter choice to get a reasonable baseline, since finetuning on those in-domain data can easily get performance improvements.**
>
> There may be a misunderstanding, and I would like to clarify. In Table 2, we train CoOp using **single-domain** few-shot labeled data. For example, CoOp(Q) represents CoOp model trained using data from Quickdraw domain. As shown in Table 2, CoOp(Q) performs better than vallina CLIP and our UMFC in the Quickdraw domain, but significantly worse in the other 5 domains. This is because single-domain supervised fine-tuning tends to overfit to the specific training domain and cannot generalize well to other new domains.
>
> **W3: What's the extra computation cost involved in this approach and how it compares to other methods? Since the proposed approach uses test-time adaptation, I believe the authors should tell us the computation cost.**
>
> Here we report the computational cost of UMFC and other comparison methods under different scenarios. For analyzing computational cost, we report the training time, inference time and memory.
>
> - **Unsupervised Calibration:** In this scenario, the entire unlabeled training set is provided for training. Corresponding computation cost comparisons are shown in below Table. **Firstly**, UMFC incurs minimal training and inference overhead compared to CLIP.   This is because UMFC only requires a single forward pass to extract features and then calculate statistics for feature calibration. **Secondly**, when compared to few-shot fine-tuning methods like CoOp, UMFC also demonstrates lower consumption of computational resources and time.
>
>
>   | Method           | Training Time            | Inference Time | Epoch | Memory  |
>   | ---------------- | ------------------------ | -------------- | ----- | ------- |
>   | CLIP   | -                        | 86 seconds     | -     | 1797MiB  |
>   | MUST   | 10 hours (2 GPUs)        | 92 seconds     | 30    | 25944MiB |
>   | UMFC             | 2.3 seconds + 55 seconds | 86 seconds     | -     | 1887MiB |
>   | CoOp (6*1 shot)  | 32 minutes               | 83 seconds     | 50    | 7007MiB |
>   | CoOp (6*4 shots) | 160 minutes              | 83 seconds     | 100   | 7007MiB |
>
> - **Test-time Adaptation:** In this scenario, no training data is provided and the test data arrives in batches. Corresponding computation cost comparisons are shown in below Table. As seen, UMFC requires less memory than TPT and shows greater computational efficiency. Specifically, UMFC takes only 296 seconds, whereas TPT requires nearly 197 minutes.  This is because TPT requires fine-tuning the text prompt for each test sample and augmenting each test sample 64 times to ensure the reliability of the fine-tuning results, which significantly slows down TPT's inference speed.
>
>   | TTA  | Inference Time | Memory  |
>   | ---- | -------------- | ------- |
>   | UMFC | 296 seconds    | 1790MiB |
>   | TPT  | 197 minutes    | 6872MiB |

---

> > ### Author Response · Authors · 2024-08-12
> > **Look forward to your response**
> >
> > Dear Reviewer jj75,
> >
> > As the rebuttal period is ending soon, please let us know whether your concerns have been addressed or not, and if there are any further questions.
> >
> > Thanks, Authors.

---

### Official Review · Reviewer_cRCJ · 2024-07-12

**Soundness:** 2
**Presentation:** 2
**Contribution:** 2
**Rating:** 3
**Confidence:** 4

**Summary:**

This paper identifies the inherent model bias within CLIP, notably in visual and textual encoders. To mitigate the bias, authors propose a feature calibration method, termed Unsupervised Multi-domain Feature Calibration. Experiments in the setting of transductive learning and test-time adaptation show the effectiveness of the proposed method.

**Strengths:**

1. This paper is well-structured and easy to follow.
2. The motivation of this paper is clear.

**Weaknesses:**

1. The proposed method is inspired by the observation in the TSNE figure (Figure 1(b)) that "features from the same... different domains". However, this does not appear to be the case. Features from QuickDraw, Painting, and Infograph are clustered, while features from Real, Clipart, and Sketch are dispersed throughout. This suggests that CLIP models can encode some domain information, but this ability is limited to specific domains. If the observation which inspires the method does not hold, the rationale for the proposed method becomes less convincing.

2. For figure 1(c), it is suggested to show the percentage of predictions rather than the absolute numbers. Each domain in DomainNet has different number of samples in validation set. Comparing the number of predictions is less convincing to show the bias of textual encoders. In addition, only Painting and Quickdraw are shown. It would be interesting to evaluate such bias in other domains.

3. It is important to reveal the process of hyper-parameter selection. The cluster number is set to 6 for DomainNet, which is equal to the number of domains in DomainNet. Does it mean that the proposed method need prior knowledge on the number of domains in the target dataset? Furthermore, it is also valuable to reveal the hyper-parameter in ImageNet-variant experiments.

4. It has been show that pre-training dataset distribution determines the CLIP's robustness [A]. Given that the proposed method is bases on CLIP bias, it would be interesting to know whether the observations persist when the pre-training dataset distribution changes.

5. The compared baselines are out-dated. Please include methods such as MaPLe[B] and PromptSRC[C] and even more recent methods.

6. It is unsure whether the comparison to other methods is fair. In UC setup, full unlabelled training set is provided, while CoOp and CLIP-Adapter are only given roughly two thousands images.

     a) Though labelling images can be expensive, collecting this amount of unlabelled data for clustering is even more expensive. Note that, these data all belong to 345 classes defined in DomainNet with six different styles.

     b) In Table 2, it is also unfair to show the superiority of UMFC by comparing it with CoOp trained on one single domain, given that UMFC is able to see images with different styles.

     c) It can be beneficial to compare the clustering time of UMFC with CoOp training time.

7. In Line 228, it is union or intersection of class sets in ImageNet-A and ImageNet-R?

[A] Data Determines Distributional Robustness in Contrastive Language-Image Pre-training (CLIP)

[B] Maple: Multi-modal prompt learning.

[C] Self-regulating Prompts: Foundational Model Adaptation without Forgetting.

**Questions:**

See weakness.

**Limitations:**

No clear negative societal impacts.

---

> ### Author Rebuttal · Authors · 2024-08-07
>
> **W1: CLIP's ability to encode domain information is limited to specific domains.**
>
> This is a valuable question. We would like to clarify that the feature bias of CLIP is **not** limited to some specific domains, but is a general issue.
> - **Explanation of Figure 1(b):** We use t-SNE for visualization, which maps high-dimensional data into lower-dimensional space while preserving relative distances. Large distances between I/P/Q  and other domains cause the relative distance among C, R, and S domains to shrink. To mitigate this effect, we visualize only C, R and S domains, As shown in Figure 1(a) of global response PDF, image features from R, C, and S domains are clustered rather than dispersed.
> - **Quantitative results of model bias:** To further illustrate model bias, we calculate the Euclidean distances of image features across 6 domains in DomainNet.
> See the detailed results and analysis in global response R2.
> - **Model bias in other datasets:** We utilize the image features from ImageNet-Variants, then visualize clustered features with t-SNE (Figure 1(b) of global response PDF) and report the Euclidean distances. These results validate the model bias in CLIP is a general issue.
>
> **W2: For figure 1(c), it is suggested to show the percentage of predictions rather than the absolute numbers.  It would be interesting to evaluate such bias in other domains.**
>
> (1) We have redrawn the figure,  shown in Figure 2 of global response PDF.
> (2) The Table 1 of global response PDF lists the top-5 predicted classes for each domain, shown as "class name / percentage". Note that a uniform prediction probability in DomainNet is 1/345 = 0.29%. The table reveals that CLIP favors different classes across domains, which is consistent with conclusion of our paper.
>
> **W3: Does it mean that the proposed method need prior knowledge on the number of domains in the target dataset?**
>
> Our method does not need prior knowledge on the number of domains in the target dataset. See the ablation studies in global response R3.
>
> **W4: Whether the observations persist when the pre-training dataset distribution changes.**
>
> To answer this question, we use pre-trained models from OpenCLIP[1], which explores scaling laws with LAION dataset. We visualize the image features on DomainNet with different pre-training data distribution. See the details and analysis in global response R2.
>
> Moreother, in Table 2 of global response PDF, we futher verify the effectiveness of our method on OpenCLIP series models. The results demonstrate the universal applicability of our method to various VL models with different architectures and pre-training data.
>
> > [1] Reproducible scaling laws for contrastive language-image learning. CVPR23
>
> **W5: Compared with more recent methods, such as MaPLe[B] and PromptSRC[C].**
>
> We compare UMFC with MaPLe and PromptSRC under different scenarios. We will add these results in final version.
> - **Domain-balanced Labeled data:** We first consider the domain-balanced data for FSL methods. Specifically, we train MaPLe and PromptSRC with k (k=1 or 4) labeled data per class for each domain on DomainNet.   The Table 3(a) of global response PDF shows that UMFC achieves comparable performance to MaPLe with limited labeled data, but underperforms them when more labeled data is available.
>
> - **Domain-imbalanced Labeled data:** We next consider domain-imbalanced labeled data, as balanced annotations in multi-domain scenarios are expensive. Specifically, we train FSL methods with 1 labeled data per class from a single domain and test them on all domains.
> The Table 3(b) of global response PDF shows that the choice of training domain significantly impacts the performance of two FSL methods, suggesting that supervised FSL methods may overfit to the training domain.
>
> In summary, FSL methods require domain-balanced labeled data for fine-tuning.
> When such high-quality labeled data is expensive to acquire in multi-domain scenarios, UMFC offers a cost-effective alternative, as collecting unlabeled data is more economical.
>
> **W6.1: Collecting unlabelled data for clustering is even more expensive than labeled data.**
>
> We need to highlight that collecting unlabeled data is more cost-efficient than labeled data, especially in multi-domain scenarios.
> - **No Need for Expert Annotation:** Unlabeled data collection avoids the need for expert annotation across domains and categories, unlike labeled data, which requires expert involvement.
> - **Challenges with Labeled Data Quality:** Although FSL methods like CoOp and CLIP-Adapter require only few labeled data, they demand high-quality data. Issues such as category imbalance, domain imbalance (shown in Table 2), and data noise can easily cause models overfitting to fine-tuned data, making labeled data collection costly in multi-domain scenarios.
>
> **W6.2: In Table 2, it is unfair to show the superiority of UMFC by comparing it with CoOp trained on one single domain.**
>
> Table1 compares UMFC with FSL methods with multi-domain training data, and Table2 use single-domain data. These experiments show UMFC as a viable alternative when collecting extensive multi-domain (balanced) labeled data for FSL methods is impractical. UMFC also generalizes well to unseen domains rather than overfitting to training domains.
> See our response to Weakness 2 for Reviewer Xgmy for detailed results.
>
> **W6.3: It can be beneficial to compare the clustering time of UMFC with CoOp training time.**
>
> We report the computational cost of UMFC and CoOp on DomainNet. UMFC's clustering time is negligible (2.3 seconds). The entire training process is significantly faster than CoOp.  UMFC does not require gradient back-propagation, leading to lower memory usage. See more details in our response to Weakness 3 for Reviewer jj75.
>
> **W7: In Line 228, it is union or intersection of class sets in ImageNet-A and ImageNet-R?**
>
> The class space of ImageNet-Variants is **union** of class sets in ImageNet-A and ImageNet-R.

---

> ### Author Response · Authors · 2024-08-12
> **Look forward to your response**
>
> Dear Reviewer cRCJ,
>
> As the rebuttal period is ending soon, please let us know whether your concerns have been addressed or not, and if there are any further questions.
>
> Thanks, Authors.

---

> > ### Comment · Reviewer_cRCJ · 2024-08-12
> >
> > Thanks authors for the rebuttal. However, my concerns are not fully addressed, so that I maintain my score as 3.
> >
> > 1) Even only plotting, Domain-C, R, S, it can still be observed that Clipart and Sketch are not well separated. They are perceptually different domains but CLIP cannot encode them well, which means the observations do not hold.
> >
> > 2) Figure 1(c) is used to show that "CLIP tends to classify images into categories whose name are closely related to corresponding domain". However, after showing the results on more domains, this claim is not well-supported, especially on DomainNet-R.
> >
> > 3) With the pre-training dataset distribution changes, the features from some domains are not separated from the others. The improvement by the proposed method is also very limited (the performance on some domains even **declines**).
> >
> > 4) The response to W6.1 is not convincing. Authors implements K-means clustering on DomainNet training set. All of these images are from 345 classes with six different styles and high quality. It has not been shown that clustering on a random web-crawled dataset brings the same benefits. Suppose that a new task comes, to generate a high quality cluster, some experts are actually needed to gather images from some specific classes with different styles and high quality. This process is in fact more expensive than labelling few shots examples from each class.
> >
> > 5) For computational cost, it seems that the feature extraction time is not considered. May I also know if it is possible to reveal which package you use for clustering? I personally also clustered features from DomainNet before, but it seems not as fast as 2.3 seconds.
> >
> > 6) For W6.2, UMFL clusters features from six domains, while CoOp is trained on only one domain. This comparison is not fair.

---

> > > ### Author Response · Authors · 2024-08-13
> > > **Response to the comment 1-3**
> > >
> > > **C1.Even only plotting, Domain-C, R, S, it can still be observed that Clipart and Sketch are not well separated. They are perceptually different domains but CLIP cannot encode them well, which means the observations do not hold.**
> > >
> > > For Figure 1(a) of global response PDF, the reviewer thinks Clipart and Sketch are not well separated, thus CLIP cannot encode perceptually different domains. However, this observation and conclusion are both incorrect.
> > >
> > > **Firstly**, Figure 1(a) shows that the image features from Clipart, Sketch and Real domains are mainly clustered in the left, middle, and right regions of the t-SNE space, respectively.
> > > **Secondly**, we present the quantitive results about the distance between domains in global response R2.
> > > As shown in the table, even the closest two domains (Clipart and Sketch) have a distance of 0.19, which is significantly larger than the variation within a single domain (0.002).
> > > The above results demonstrate that CLIP's visual encoder encodes different domains within the representation space.
> > >
> > > **C2.Figure 1(c) is used to show that "CLIP tends to classify images into categories whose name are closely related to corresponding domain". However, after showing the results on more domains, this claim is not well-supported, especially on DomainNet-R.**
> > >
> > > - **Explanation of DomainNet-R**
> > >
> > >   For the Table 1 of global response PDF, there may be some misunderstanding.
> > >   DomainNet [1] contains 6 domains, and DomainNet-R denotes the "Real" domain.
> > >   However, the "Real" domain lacks clear conceptual definition, thus there are no categories closely related to this domain and the text encoder bias may be not observious.
> > >   The results in Table 1 of global response PDF also verify this point, as the percentages of top-5 predictions (0.47%-0.63%) is near by the uniform prediction (0.29%).
> > >
> > > - **Text Encoder Bias**
> > >
> > >   In Lines 145-146 of our paper, we define text encoder bias as "*CLIP exhibits a preference for domain-related categories in specific domains,*" and empirically, CLIP demonstrates worse zero-shot performance in these domains.
> > >   For example, the percentages of top-1 predictions for classes in the Quickdraw and Infograph domains are 21.9% and 4.91%, respectively, which are significantly higher than the uniform prediction (0.29%).
> > >   Meanwhile, zero-shot CLIP achieves only 14.23% and 49.47% accuracy in these two domains, much lower than its performance in other domains (e.g. accuracy of 82.98% in Real domain).
> > >   We believe that CLIP's category preference is closely related to its poor performance in multi-domain scenarios.
> > >
> > > > [1] Moment Matching for Multi-Source Domain Adaptation. ICCV2019
> > >
> > >
> > > **C3.With the pre-training dataset distribution changes, the features from some domains are not separated from the others. The improvement by the proposed method is also very limited (the performance on some domains even declines).**
> > >
> > > - **Feature Visualization**
> > >
> > >   For the Figure 3(a) in the global response PDF, the reviewer thinks the domain features are not separated from the others.
> > >   However, the conclusion is incorrect.
> > >   We can observe that the image features from Clipart, Infograph, Painting, Quickdraw, Skecth are well separated.
> > >   Besides, we calculate the Euclidean distances of  image features between different domains using OpenCLIP.
> > >   As shown in the table below, even the closest two domains (Clipart and Sketch) have a distance of 0.27, which is significantly larger than the variation within a single domain (0.00195).
> > >
> > >   | DomainNet | C    | I    | P    | Q    | R    | S    |
> > >   | -------- | ---- | ---- | ---- | ---- | ---- | ---- |
> > >   | C        | 0.00 | 0.36 | 0.37 | 0.53 | 0.32 | 0.27 |
> > >   | I         | 0.36 | 0.00 | 0.44 | 0.67 | 0.36 | 0.43 |
> > >   | P         | 0.37 | 0.44 | 0.00 | 0.66 | 0.29 | 0.37 |
> > >   | Q         | 0.53 | 0.67 | 0.66 | 0.00 | 0.60 | 0.51 |
> > >   | R         | 0.32 | 0.36 | 0.29 | 0.60 | 0.00 | 0.37 |
> > >   | S         | 0.27 | 0.43 | 0.37 | 0.51 | 0.37 | 0.00 |
> > >
> > > - **Limited Improvment**
> > >
> > >   We think the improve by UMFC is not limited.
> > >   We evaluate our UMFC on a wide range of models, involving different backbones and pre-training data. Across all experiments, our method consistently improved average performance. For instance, in the Quickdraw domain, we improved performance from 15.63% to 22.61%.
> > >   The only exception was a marginal decline (0.1% ~ 0.6%) in the Real domain, while all other domains showed performance gains.
> > >   We believe that this decline can be easily addressed by adjusting the hyperparameters.

---

> > > > ### Author Response · Authors · 2024-08-13
> > > > **Response to the comment 4-6**
> > > >
> > > > **C4. The response to W6.1 is unconvincing. The authors implemented K-means clustering on DomainNet training set, which consists of high-quality images. It’s unclear if clustering on a random web-crawled dataset would yield similar benefits. For a new task, generating high-quality clusters would require experts to curate images from specific classes with different styles, which could be more costly than labeling a few examples per class.**
> > > >
> > > > - **Requirement of a high quality dataset for UMFC**
> > > >
> > > >   Our UMFC does **not** require high quality dataset with all different styles and categories for clustering. **(1)**
> > > >   In Section 5.4 of our paper, we consider the test-time adaptation scenario, where the test data arrives in batches.
> > > >   In our experiments, the batch size ranges from 1 to 100 (as detailed in our response to Question 1 for reviewer hYxj), significantly smaller than 2070 (6 domains * 345 classes), meaning each batch only contains  a subset of categories and domains.
> > > >   Despite this, our UMFC still performs well, showcasing its generalization and robustness.
> > > >   **(2)** Besides, our UMFC can also effectively handle single-domain (see our response to reviewer hYxJ's W1) and partial-domain scenarios (see our response to reviewre Xgmy's W2).
> > > >
> > > > - **Experiments on a random web-crawled dataset**
> > > >
> > > >   To best of our knowledge, DomainNet is the largest existing multi-domain dataset collected from the web for image classification. We have demonstrated that our method outperforms CLIP and performs comparably to other methods that require additional annotations or optimization in DomainNet.
> > > >   Evaluating the performance of UMFC on a more realistic, randomly web-crawled dataset, which would involve significant time and effort, lies beyond the scope of this paper.
> > > >
> > > > - **Data Collection Cost**
> > > >
> > > >   **(1)** Collecting unlabeled data is much easier than collecting labeled data, which is the motivation behind tasks like unsupervised fine-tuning [2] and Multi-target Domain Adaptation [3].
> > > >   In real-world scenarios like autonomous driving, data sources are complex and vast, making labeling particularly challenging. However, unlabeled data can be efficiently and automatically collected. In contrast, gathering labeled data requires first collecting data from various domains and categories, and then labeling it, making it much more time-consuming than collecting unlabeled data. **(2)** Furthermore, the performance of few-shot fine-tuning methods is more sensitive to the quality of the labeled data, and demond more training time and memory than UMFC (see our response to jj75's W3).
> > > >
> > > > > [2] MAtch, eXpand and Improve: Unsupervised Finetuning for Zero-Shot Action Recognition with Language Knowledge. ICCV 2023 \
> > > > > [3] Multi-Target Domain Adaptation with Collaborative Consistency Learning. CVPR21
> > > >
> > > > **C5.For computational cost, it seems that the feature extraction time is not considered. May I also know if it is possible to reveal which package you use for clustering?**
> > > >
> > > > We first use CLIP's image encoder to extract all image features (i.e., data), a process that takes only **35.67 seconds**. We utilize the K-Means algorithm from sklearn. The clustering code is provided below, where the shape of data is [32768,512]. If you're interested, you can try running it on a 4090 Ti.
> > > > ```python
> > > > def kmeans(data, n_clusters=3):
> > > >     from sklearn.cluster import KMeans
> > > >     model = KMeans(n_clusters=n_clusters, max_iter=500)
> > > >     y_pred = model.fit_predict(data)
> > > >     return y_pred
> > > > ```
> > > >
> > > > **C6.For W6.2, UMFC clusters features from six domains, while CoOp is trained on only one domain. This comparison is not fair.**
> > > >
> > > > We conduct experiments on DomainNet, where the unlabeled training data for UMFC originates from a single domain and the test data contains all six domains.
> > > > **For a fair comparison**, we provide UMFC with 8*345 unlabeled samples (each class has 8 samples) from a **single domain** as the unlabeled training data.
> > > > Note that, CoOp is trained using the same number of training data, but with labels for supervised fine-tuning.
> > > > For clarity, we use CoOp (C/Q/I) and UMFC (C/Q/I) to denote training samples for CoOp and UMFC from the Clipart/Quickdraw/Infograph domains, respectively.
> > > >
> > > > As shown in the table below, with the same amount of training data, our UMFC achieves consistent performance improvements on both training and unseen domains, while the CoOp method suffers significant performance drops on unseen domains. Additionally, unlike CoOp, our UMFC does not require labeled data or any parameter fine-tuning.
> > > >
> > > > ||C|I|P|Q|R|S|Avg|
> > > > |--------|-----|-----|-----|-----|-----|-----|-----|
> > > > |CLIP|71.21|49.47|64.61|14.23|82.98|64.81|57.88|
> > > > |CoOp (C)|74.55|42.66|55.94|13.82|75.00|58.73|53.45|
> > > > |UMFC (C)|73.27|52.96|65.27|16.94|83.60|67.04|**59.85**|
> > > > |CoOp (Q)|43.97|25.50|32.63|29.07|48.44|38.74|36.39|
> > > > |UMFC (Q)|72.17|49.65|63.85|17.47|82.84|66.36|**58.72**|
> > > > |CoOp (I)|60.19|54.28|50.81|11.19|70.73|54.27|50.24|
> > > > |UMFC (I)|72.54|55.21|64.48|16.30|83.31|66.51|**59.73**|

---

### Official Review · Reviewer_Xgmy · 2024-07-13

**Soundness:** 2
**Presentation:** 3
**Contribution:** 3
**Rating:** 6
**Confidence:** 2

**Summary:**

**I am not an expert in this domain. So my review may not be informative.**

The paper introduces Unsupervised Multi-domain Feature Calibration (UMFC), designed to improve the adaptability of Vision-Language Foundation Models like CLIP to various downstream tasks across multiple domains using unlabeled data. The authors pinpoint model biases within CLIP’s visual and textual encoders that affect its performance across different domains. To mitigate these biases, UMFC employs two calibration modules: the Image Feature Calibration (IFC) and the Text Feature Calibration (TFC). These modules recalibrate the model's focus from domain-specific to category-level information, enhancing its generalization capability. The effectiveness of UMFC is demonstrated through significant performance improvements in unsupervised calibration, transductive learning, and test-time adaptation tasks.

**Strengths:**

1. UMFC addresses the core issue of domain shift by recalibrating both visual and textual features to be domain-agnostic
2. The framework leverages the typically abundant but underutilized unlabeled data in practical scenarios, offering a cost-effective solution for enhancing model performance without the need for labeled data.
3. The effectiveness of UMFC is robustly validated across multiple downstream tasks, demonstrating its practical utility and the potential for wide application in real-world scenarios.

**Weaknesses:**

1. While effective, the calibration process might introduce additional complexity in terms of understanding and implementing the recalibration mechanisms, particularly the calculation and subtraction of domain-specific biases.
2. There's a risk that overly aggressive calibration could lead to overfitting on the specific domains included in the training set, potentially reducing the model’s ability to generalize to entirely new or unseen domains.

**Questions:**

Please see weaknesses.

**Limitations:**

None.

---

> ### Author Rebuttal · Authors · 2024-08-07
>
> **W1: While effective, the calibration process might introduce additional complexity in terms of understanding and implementing the recalibration mechanisms, particularly the calculation and subtraction of domain-specific biases.**
>
> In fact, our calibration method is straightforward and computationally efficient.
>
> - **Easy to implement:** The calibration process involves three main steps:
>   1. **Assigning cluster label:** (stage1) Unlabeled samples are clustered, and cluster labels are assigned.
>   2. **Collecting statistical information:** (stage2)  Statistics for each cluster are gathered based on the assigned labels.
>   3. **Feature calibration:** (stage3) The original image and text features are calibrated using the collected statistics.
>
> - **Computation efficient:** We utilize the k-means algorithm to cluster image features and calculate the mean feature of each cluster. Therefore, steps 1 and 2 are efficient and can be completed within one minute in our experiments. In step 3, the feature calibration mechanism does not require backpropagation or modification of model parameters, leading to minimal additional computational overhead. For further details, please refer to our response to Weakness 3 for Reviewer jj75.
>
>
> **W2: There's a risk that overly aggressive calibration could lead to overfitting on the specific domains included in the training set, potentially reducing the model’s ability to generalize to entirely new or unseen domains.**
>
> Thanks for this valuable suggestion.  To evaluate UMFC's generalization capability on unseen domains, we follow the unsupervised calibration setting in Section 5.2. We randomly select disjoint domains as unlabeled training or test data. Specifically, we randomly select 3 domains in DomainNet as unlabeled training data (source domains) and use the remaining 3 domains as test data (target domains). The results are shown in below table.
>
> From this table, we can observe that, UMFC consistently outperforms vanilla CLIP model across different source domains, demonstrating its robust generalization ability. Additionally, our method can be easily deployed in Test-Time Adaptation (TTA) scenarios. This allows for continuous updates of statistical information as new test samples from emerging domains arrive. This capability ensures that the model maintains its generalization effectiveness over new domains.
>
>
> | Method | Source Domain |     Target Domain    |  Target Domain        |      |
> |:------:|:-------------:|:--------------:|:--------------:|:----:|
> |        |               |       CQS      |       IPR      |  Avg |
> | CLIP   | -             | 71.2/14.2/64.8 | 49.5/64.6/83.0 | 57.9 |
> | UMFC   | CQS           | 73.6/20.5/67.7 | 56.4/66.2/84.2 | 61.4 |
> | UMFC   | IRP           | 73.3/17.7/67.8 | 56.2/67.5/84.3 | 61.1 |

---

> > ### Author Response · Authors · 2024-08-12
> > **Look forward to your response**
> >
> > Dear Reviewer Xgmy,
> >
> > As the rebuttal period is ending soon, please let us know whether your concerns have been addressed or not, and if there are any further questions.
> >
> > Thanks, Authors.

---

> > > ### Comment · Reviewer_Xgmy · 2024-08-13
> > >
> > > Thank you for the detailed responses. My concerns have been addressed, and I would like to update my score to 6.

---

### Official Review · Reviewer_HbLd · 2024-07-17

**Soundness:** 3
**Presentation:** 2
**Contribution:** 3
**Rating:** 6
**Confidence:** 3

**Summary:**

This paper attempts to address the problem of domain gap in existing VLMs. Based on an unlabelled mixed dataset, this paper first proposes to fit each different domain and remove the domain bias of image features by Gaussian mixture model. Furthermore, the domain bias of text features is similarly removed by the consistency of image and language modalities. The method is validated on several different tasks.

**Strengths:**

The methodology is simple and easy to understand, and the experiments are relatively extensive.

**Weaknesses:**

The proposed idea is simple and reasonable. My major concern is whether the proposed methodology is a real improvement over CLIP-D (e.g., 61.68 vs. 61.03 in unsupervised calibration), considering the requirement for extra computational overhead and unlabeled dataset.

0. how does it compare to CLIP-DE (Domain-Specific Ensemble Prompting)?
1. the Gaussian assumption is a bit too strong for each domain.
2. I would assume the dataset to be unlabelled, as described in L179-L180. However, for text feature calibration, the description in L208 is _'By clustering image features, we can compute the average feature $µ_i$ of unlabelled images in each domain $i$.
representing the domain-specific features of that domain'_ . Does the domain here refer to the clustering in the image feature calibration step?
3. Equation (3) should be $\triangleq$ .
4. More details are needed for each experimental section.

**Questions:**

See weakness.

---

> ### Author Rebuttal · Authors · 2024-08-06
>
> **My major concern is whether the proposed methodology is a real improvement over CLIP-D (e.g., 61.68 vs. 61.03 in unsupervised calibration), considering the requirement for extra computational overhead and unlabeled dataset.**
>
> Thanks for this question, and I would like to clarify this point in the following.
>
> - **Improvement over CLIP-D.** In CLIP-D, we design customized prompts that incorporates the corresponding domain label and name for each test sample.
> Compared to vallina CLIP, CLIP-D shows better performance, which demonstrates the benefit of considering domain information under multi-domain scenarios.
> However, our comparison between UMFC and CLIP-D is not intended to show that UMFC performs better than CLIP-D, as CLIP-D utilizes domain labels and names for all test samples, while UMFC does not.
> Instead, this comparison aims to illustrate that UMFC can extract domain information from data **without supervision**, making it more versatile in various situations.
>
> - **Disadvantages of CLIP-D.** Although CLIP-D a is simple and effective method in multi-domain scenarios, it has many disadvantages: 1. it requires domain labels and names for all test samples. 2. it is sensitive to the choice of domain names. See more details in our global response R1.
>
> **W0: How does UMFC compare to CLIP-DE (Domain-Specific Ensemble Prompting)?**
>
> We build CLIP-DE by replacing the single domain-specific prompt in CLIP-D with an ensemble of prompt template with domain names. As shown in the table below, CLIP-DE performs only marginally better than CLIP-E, and slightly worse than CLIP-D on DomainNet. We analyze this is because the ensemble strategy weakens the domain information utilized in CLIP-D. Furthermore, our UMFC outperforms CLIP-DE, validating its superiority.
>
> | Method     | C | I | P | Q| R | S | Avg |
> |------------|-----------|----------|----------|----------|----------|----------|----------|
> | CLIP-E | 73.16 | 54.17 | 67.02 | 15.86 | 84.30 | 67.49 | 60.33 |
> | CLIP-D | 73.90 | 55.84 | 67.75 | 17.84 | 83.26 | 67.56 | 61.03 |
> | CLIP-DE| 73.62 | 54.34 | 67.64 | 16.34 | 84.49 | 67.45 | 60.65 |
> | CLIP-E+UMFC       | 73.84 | 56.59 | 67.39 | 20.03 | 84.33 | 67.9  | **61.68** |
>
>
>
> **W1: Gaussian assumption is a bit too strong for each domain.**
>
> Thanks for your suggestions. In domain generalization, it is commonly assumed that domain features follow a  multivariate Gaussian distribution [1]. Moreover, since our method relies only on the mean values for feature calibration (Equation 4 of main paper), it can generalize to other distributions where the mean effectively captures the overall information.
>
> > [1] Domain Generalization With Adversarial Feature Learning. CVPR 2018
>
>
> **W2: I would assume the dataset to be unlabelled, as described in L179-L180. However, for text feature calibration, the description in L208 is *'By clustering image features, we can compute the average feature µ of unlabelled images in each domain . representing the domain-specific features of that domain'* . Does the domain here refer to the clustering in the image feature calibration step?**
>
> Yes, we apologize for this confusion. Here, 'domain' refers to the cluster labels assigned during the clustering stage. We will revise it in the final version.
>
> **W3: Equation (3) should be $\triangleq$.**
>
> Thank you for pointing this! We will revise it in final version.
>
>
> **W4: More details are needed for each experimental section.**
>
>   Thanks for your suggestions. Our work can be deployed across various scenarios, including Unsupervised Calibration (UC), Test-Time Adaptation (TTA), and Transductive Learning (TL).  We will add the following details for the experimental section in final version.
>
>    + In the UC scenario, we provide an unsupervised training set and use K-Means to assign cluster labels to the training samples, while also saving the corresponding cluster prototypes. Then, we create an unlabeled training set from a mixed domain by sampling 16 instances from each class across 6 domains in DomainNet. UMFC derives image bias and text bias for different clusters based on the cluster labels. During the testing phase, we first predict the cluster labels of the test samples using the cluster prototypes obtained during training, and then calibrate the predictions using the bias information derived from UMFC.
>
>    + In the TTA scenario, no training data is provided. Test data from mixed domains arrive in batches, with a batch size set to 100. For the first batch, we perform initial clustering using K-Means. For subsequent batches, we assign cluster labels to the samples based on cluster prototypes and continuously update these prototypes. Once the cluster labels are obtained, UMFC calculates the bias information for the current batch, updates the bias information for each cluster based on the new labels, and then calibrates the data for the current batch.
>
>    + In the TL scenario, we provide the entire test set. Similar to UMFC, we gather statistical information and calibrate the predictions for the test data. TL can be viewed as an extreme case of TTA, where the entire test set is treated as a single batch.

---

> > ### Comment · Reviewer_HbLd · 2024-08-11
> >
> > Thanks for the response. Could the author supplement more details of CLIP-DE method? As I would expect it to be at least as good as CLIP-D/CLIP-E.

---

> > > ### Author Response · Authors · 2024-08-11
> > > **Response to the comment**
> > >
> > > **Could the author supplement more details of CLIP-DE method? As I would expect it to be at least as good as CLIP-D/CLIP-E.**
> > >
> > > Thank you for your question. In the following, we provide more details about CLIP-E, CLIP-D and CLIP-DE in DomainNet dataset.
> > >
> > > - CLIP-E
> > >
> > >   In CLIP-E, we use an ensemble of 7 prompts from the following list.
> > >   ```python
> > >   prompts = [
> > >       "itap of a {class}.",
> > >       "a bad photo of the {class}.",
> > >       "a origami {class}.",
> > >       "a photo of the large {class}.",
> > >       "a {class} in a video game.",
> > >       "art of the {class}.",
> > >       "a photo of the small {class}.",
> > >   ]
> > >   ```
> > >
> > > - CLIP-D
> > >
> > >   Different with CLIP-E, CLIP-D utilizes a single prompt but with domain information from the test samples. The prompt used in CLIP-D for each test sample is
> > >   ```python
> > >   prompt = ["a {domain} image of a {class}."]
> > >   ```
> > >   Here, {domain} represents the domain name of the test sample, which belongs to one of the six domains in DomainNet: clipart, infograph, painting, quickdraw, real, or sketch.
> > >
> > > - CLIP-DE
> > >
> > >   Finally, we build CLIP-DE by replacing the single domain-specific prompt in CLIP-D with an ensemble of 8 prompt templates with domain names.
> > >   Specifically, these templates include the 7 prompts from CLIP-E and 1 domain-specific prompt from CLIP-D. The prompt list for CLIP-DE is:
> > >   ```python
> > >   prompts = [
> > >       "itap of a {domain} {class}.",
> > >       "a bad photo of the {domain} {class}.",
> > >       "a origami {domain} {class}.",
> > >       "a photo of the large {domain} {class}.",
> > >       "a {domain} in a {class} video game.",
> > >       "art of the {domain} {class}.",
> > >       "a photo of the small {domain} {class}.",
> > >       "a {domain} image of a {class}."
> > >   ]
> > >   ```
> > >
> > >   The table in our response to W0 shows that CLIP-DE outperforms CLIP-E, indicating that domain information is indeed beneficial. However, CLIP-DE performs slightly worse than CLIP-D. We attribute this to the ensemble strategy, which weakens the domain information utilized in CLIP-D.
> > >   This hypothesis is supported by the some experimental observations.
> > >   As shown in the table below, using the same domain information with different prompt templates can significantly impact zero-shot performance.
> > >   We think this is due to variations in prompt design, which may reduce the effectiveness of domain-specific cues, leading to suboptimal results in some cases.
> > >
> > >   | prompt                                | C     | I     | P     | Q     | R     | S     | Avg   |
> > >   | ------------------------------------- | ----- | ----- | ----- | ----- | ----- | ----- | ----- |
> > >   | "a {domain} of a {class}."            | 73.43 | 55.62 | 67.37 | 17.29 | 82.76 | 66.60 | 60.51 |
> > >   | "a photo of a {class} from {domain}." | 73.31 | 54.88 | 67.04 | 15.81 | 82.76 | 66.51 | 60.05 |
> > >   | "a {domain} image of a {class}."      | 73.90 | 55.84 | 67.75 | 17.84 | 83.26 | 67.56 | 61.03 |

---

> > > > ### Comment · Reviewer_HbLd · 2024-08-12
> > > >
> > > > Thanks for the response. In my experience, the prompting template matters a lot in CLIP zero-shot classification, even a comma will affect the accuracy. I want to ensure the improvement do come from the proposed idea. Considering the limited time for the author to response, could you please do the 6x7 style prompts? By concatenating the two prompts from CLIP-D and CLIP-E.

---

> > > > > ### Author Response · Authors · 2024-08-12
> > > > > **Response to the comment**
> > > > >
> > > > > **Could you please do the 6x7 style prompts? By concatenating the two prompts from CLIP-D and CLIP-E.**
> > > > >
> > > > > Thank you for your response. Based on your suggestion, we concatenated the two sets of prompts from CLIP-D and CLIP-E. For each domain, we created 7 prompts, resulting in a total of 42 prompts (6 domains * 7 prompts) for each class.
> > > > >
> > > > > ```python
> > > > > prompts=[
> > > > >   "a {domain} image of a {class}. itap of a {class}.",
> > > > >   "a {domain} image of a {class}. a bad photo of the {class}.",
> > > > >   "a {domain} image of a {class}. a origami {class}.",
> > > > >   "a {domain} image of a {class}. a photo of the large {class}.",
> > > > >   "a {domain} image of a {class}. a {class} in a video game.",
> > > > >   "a {domain} image of a {class}. art of the {class}.",
> > > > >   "a {domain} image of a {class}. a photo of the small {class}."
> > > > > ]
> > > > > ```
> > > > >
> > > > > In the table below, we still observe that the ensemble strategy underperforms compared to original CLIP-D, suggesting that integrating all 42 prompts across 6 domains may weaken domain-specific information.
> > > > >
> > > > > | Method     | C     | I     | P     | Q     | R     | S     | Avg   |
> > > > > | ---------------------- | ----- | ----- | ----- | ----- | ----- | ----- | ----- |
> > > > > | CLIP-D                    | 73.90 | 55.84 | 67.75 | 17.84 | 83.26 | 67.56 | **61.03** |
> > > > > | Ensemble (6 domains * 7 prompts)                     | 72.46 | 54.41 | 66.75 | 15.06 | 84.33 | 66.87 | 59.98 |

---

> > > > > > ### Comment · Reviewer_HbLd · 2024-08-12
> > > > > >
> > > > > > Many thanks for the new results. Although, the new results still fall short of the my expectations, requesting further experiments may be unreasonable due to time constraints. The reviewer believes there should be a more suitable strategy for integrating CLIP-D and CLIP-E, and the reviewer still believes that the ensembling approach should achieve results at least as good as those of CLIP-D/CLIP-E.
> > > > > >
> > > > > > Nevertheless, I will increase the rating to a 6 - weak accept. As far as I know (may not be complete), previous work on improving CLIP's zero-shot classification ability has typically focused solely on enhancing the template itself. Despite some remaining issues with model evaluation aspect, the focus on domain information in this paper is valuable to the community.

---

### Author Rebuttal · Authors · 2024-08-07

### **R1. Disadvantages of CLIP-D.**
CLIP-D serves as a comparison baseline by incorporating the domain names of test samples into its prompts. While CLIP-D demonstrates better performance compared to the CLIP model in Table 1, it has several notable drawbacks:

- **Dependency on Domain Labels and Names**: CLIP-D requires domain labels and names for all test samples, which is impractical in many real-world scenarios. In contrast, our UMFC method estimates the image feature bias from clustered features, without the need for domain labels and names of the test samples.

- **Sensitivity to Domain Name Selection**: CLIP-D's performance is sensitive to the choice of domain names. As shown in the table below, CLIP-D's effectiveness degrades when synonyms are substituted for domain names.

   |                   | C     | I     | P     | Q     | R     | S     | Avg   |
     | ----------------- | ----- | ----- | ----- | ----- | ----- | ----- | ----- |
     | CLIP-D            | 73.90 | 55.84 | 67.75 | 17.84 | 83.26 | 67.56 | 61.03 |
     | CLIP-D (Synonyms) | 72.11 | 54.83 | 66.09 | 17.73 | 83.34 | 65.74 | 59.97 |

In summary, CLIP-D relies on extensive prior knowledge to perform better in the multi-domain scenarios.
In contrast, UMFC's calibration process does not require domain labels and names of test samples, enabling it to handle a broader range of real-world scenarios.

### **R2. Generality of our observation.**
Here we would like to emphasize that our observation (in Figure 1) is a general issue for CLIP model.

- **Quantitative results of model bias:** To further quantify model bias of CLIP, we calculate the Euclidean distances of image features across different domains in DomainNet and ImageNet-Variants.
The tables below show that these distances (except the diagonal element) range from 0.2 to 0.6, indicating that CLIP's visual encoder incorporates domain information into image features across different domains and datasets.

  | DomainNet | C    | I    | P    | Q    | R    | S    |
  | -------- | ---- | ---- | ---- | ---- | ---- | ---- |
  | C        | 0.00 | 0.35 | 0.30 | 0.41 | 0.26 | 0.19 |
  | I        | 0.35 | 0.00 | 0.37 | 0.59 | 0.30 | 0.36 |
  | P        | 0.30 | 0.37 | 0.00 | 0.52 | 0.26 | 0.26 |
  | Q        | 0.41 | 0.59 | 0.52 | 0.00 | 0.52 | 0.38 |
  | R        | 0.26 | 0.30 | 0.26 | 0.52 | 0.00 | 0.28 |
  | S        | 0.19 | 0.36 | 0.26 | 0.38 | 0.28 | 0.00 |

   | ImageNet-Variants | IN-A | IN-R | IN-S |
   | ------------ | ---- | ---- | ---- |
   | IN-A         | 0.00 | 0.28 | 0.36 |
   | IN-R         | 0.28 | 0.00 | 0.19 |
   | IN-S         | 0.36 | 0.19 | 0.00 |

- **Different pre-training distribution:** It is an interesting question that whether our observation presist if pre-training data distribution of CLIP changes. To answer this question, we use pre-trained models from OpenCLIP[1], which explores scaling laws with public LAION dataset. We visualize image features from visual encoders on DomainNet using t-SNE, shown in Figure 3 of global response PDF. Figure 3(a) reveals that images from different domains still cluster together despite changes in pre-training data (LAION-80M, LAION-400M and LAION-2B).

### **R3. The ablation of the number of clusters M.**
We conducte additional ablation experiments on the number of clusters M and demonstrate that our method is not sensitive to the choice of this hyperparameter.
We test different values for the hyperparameter M on DomainNet and ImageNet-Variants. As shown in the table below, our UMFC consistently outperforms vanilla CLIP, even when the number of clusters M does not match the number of domains (6 for DomainNet and 3 for ImageNet-Variants). For instance, setting M from 3 to 10 all leads to improvements on DomainNet.

| DomainNet   | C     | I     | P     | Q     | R     | S     | Avg   |
| ----------- | ----- | ----- | ----- | ----- | ----- | ----- | ----- |
| CLIP        | 71.21 | 49.47 | 64.61 | 14.23 | 82.98 | 64.81 | 57.88 |
| UMFC (M=3)  | 72.53 | 54.60 | 66.31 | 20.32 | 83.35 | 66.86 | 60.66 |
| UMFC (M=4)  | 73.55 | 56.36 | 67.19 | 20.62 | 84.13 | 67.69 | 61.59 |
| UMFC (M=6)  | 73.01 | 55.44 | 66.89 | 20.14 | 83.66 | 67.51 | 61.11 |
| UMFC (M=8)  | 73.50 | 56.58 | 67.53 | 20.64 | 84.06 | 67.92 | 61.71 |
| UMFC (M=10) | 73.63 | 56.87 | 67.81 | 20.23 | 84.20 | 67.87 | 61.77 |

| IN-Variants | IN-A  | IN-R  | IN-S  | Avg   |
| ----------- | ----- | ----- | ----- | ----- |
| CLIP        | 42.13 | 66.95 | 74.58 | 61.22 |
| UMFC (M=2)  | 45.35 | 71.71 | 77.37 | 64.81 |
| UMFC (M=3)  | 44.77 | 72.19 | 78.62 | 65.19 |
| UMFC (M=6)  | 45.29 | 71.33 | 77.59 | 64.74 |

---

### Decision · Program_Chairs · 2024-09-25

**Decision:**

Accept (poster)

**Comment:**

This paper studies how to calibrate the pretrained VLM features for downstream task domains. A key observation is that the pretrained VLM features tend to form clusters according to task domains instead of semantic concepts, which can harm the downstream task performance in differentiating semantic categories. The authors propose a simple mean-shift technique to calibrate the vision features by using unlabeled images from the downstream domain, and adjust the text features in the same direction without using the knowledge of the downstream task vocabulary. This technique relies solely on images and obviates the need for downstream domain name or knowledge of the samples.

This paper receives mixed reviews with scores 6,6,6,4,3. Many reviewers appreciate the clarify and simplicity of the method (HbLd, jj75), the motivation by observing the domain-sensitivity and bias in CLIP (HbLd, jj75, hYxJ), the ability of the method to leverage unlabeled data (Xgmy), and the comprehensive experiments (HbLd, Xgmy, hYxJ). However, the reviewers also noted concerns about the paper including whether there's a real improvement over CLIP-D/E variants (HbLd, jj75), complexity and overfitting (Xgmy), fair comparison with TTA (hYxJ), validity of domain clusters in T-SNE (cRCJ), comparison with MaPLe and PromptSRC (cRCJ). The author's rebuttal was helpful to address the concerns of reviewers HbLd, Xgmy and hYxJ, but not cRCJ. To understand cRCJ's main concern about t-SNE clusters, I have examined the t-SNE visualization and quantitative measures of intra- and inter-cluster distances in the author feedback, and believed that the authors have addressed this adequately. Even though we did not hear back from reviewer jj75, the main question about CLIP-D is similar to HbLd which I think is well-addressed in the author feedback too. Compared to adaptation or FT, the proposed method is simpler because it is training-free, label-free, and can work in the black-box settings without model weight access.

Taken altogether, I'd recommend acceptance of the paper since the reasons to accept outweigh the reasons to reject in my opinion, but I think it'd be best for us to discuss this one since 2 out of 5 reviewers are leaning reject.